# The effects of temperature on nestling growth in a songbird depend on developmental constraints

Sage A. Madden [1]*, Rebecca J. Safran[2], Gail L. Patricelli[1], Sara R. Garcia[2],
Zachary M. Laubach[2]

**1** Department of Evolution and Ecology, University of California Davis, Davis, California, United States
of America, **2** Department of Ecology and Evolutionary Biology, University of Colorado Boulder, Boulder,
Colorado, United States of America

* saamadden@ucdavis.edu

## Abstract

Climate change can adversely impact animals, especially those that cannot independently thermoregulate or avoid exposure to temperature variation. Altricial bird nestlings may be particularly vulnerable to temperature variation, with cold, hot, and variable temperatures leading to reduced nestling growth. At a broad scale, temperature effects on nestling growth vary across climatic zones, but how temperature effects vary with early-life developmental constraints imposed by the timing of thermoregulatory development, competition with siblings, and the amount of parental care has received less attention. We investigated whether the effects of temperature on the body mass of wild barn swallow (*Hirundo rustica erythrogaster*) nestlings (n = 113 nestlings, 31 nests) in Boulder County, CO depends on timing of exposure during development, relative size within the brood, or level of parental feeding. Lower minimum temperatures were associated with lower nestling mass in early but not late development (before but not after the putative development of thermoregulatory independence). Additionally, we found marginal evidence that the smallest nestling in the brood was more vulnerable to extreme and variable temperatures than other brood mates. Similarly, we found evidence that cold temperatures had stronger negative effects on nestlings in nests receiving low versus high levels of parent feeding. These findings indicate the existence of fine-scale heterogeneity in which the effects of temperature on nestling development are sensitive to metabolic constraints and early-life social environment, providing insight into the factors that may provide resilience to or exacerbate temperature effects on individual birds.

## Introduction

Rising and increasingly variable temperatures due to climate change [1] can adversely impact animals, especially those that cannot independently thermoregulate or avoid exposure to temperature variation. At-risk individuals include young animals

**Data availability statement:** All data and analysis code used in this article are available in a Zenodo repository (https://doi.org/10.5281/zenodo.19077502).

**Funding:** ZML was supported by the National Science Foundation Division of Biological Infrastructure (https://www.nsf.gov/bio/dbi) 2010607. SAM was supported by a Graduate Group in Ecology Fellowship through the University of California, Davis (https://www.ucdavis.edu/) and by a National Science Foundation Graduate Research Fellowship (https://www.nsf.gov/funding/opportunities/grfp-nsf-graduate-research-fellowship-program) 2036201. Research in this paper was supported by National Science Foundation Division of Integrative Organismal Systems (https://www.nsf.gov/bio/ios) 1856266 to RJS. The funders had no role in study design, data collection and analysis, decision to publish, or preparation of the manuscript.

**Competing interests:** The authors have declared that no competing interests exist.

lacking the mechanisms required to regulate exposure and cope with temperature extremes [2]. Exposure to excessively hot or cold temperatures is a form of developmental stress that negatively impacts not only the immediate survival of young animals in many species, but also the development of adult phenotypes that affect later survival and reproduction [3,4]. Altricial bird nestlings may be particularly vulnerable to extreme and variable temperatures because they are confined to a nest, ectothermic and mostly featherless early in development [5,6], and entirely dependent on parental care [7,8]. As a result, exposure to hot, cold, or variable temperatures may impede nestling growth through multiple mechanisms, including direct impacts on metabolic costs and indirect effects via reduced food delivery by parent birds [9,10]. Although previous studies have revealed that temperature effects on nestling growth vary across climate zones [9,10], we know little about how these effects may vary with localized early-life developmental constraints imposed by the timing of thermoregulatory development, competition with brood mates, and amount of parental care. This knowledge would help identify sensitive periods of development, when altricial nestlings are at greatest risk to extreme or variable temperatures, as well as the factors that may confer resilience to or exacerbate the impacts of human-induced environmental change.

The effects of temperature on altricial nestling development may differ depending on the timing of exposure during development [11–14]. During the first few days after hatch, nestlings are mostly featherless and lack the ability to thermoregulate independently [8,15]. Consequently, exposure to hot or cold ambient temperatures may drive body temperatures outside of the narrow range optimal for growth [5,6,11,13]. Endothermy typically develops over several days in the middle of the nestling period [6,15], with timing influenced by growth rate of the species, brood size, and a variety of other factors [5,15,16]. After endothermy develops, nestlings exposed to hot or cold temperatures may expend more energy on thermoregulation and/or enter a state of hyperthermia (in the case of extreme heat), leading to increased stress and decreased growth [6,17,18]. Because the thermoregulatory abilities of nestlings change over the course of the nestling period, we expect the effects of temperature on nestling growth to vary across ontogeny. This possibility warrants further investigation, as most previous studies have focused on the nestling period as a whole (but see [11–14]).

A nestling's social environment, including size relative to brood mates, also may mitigate or exacerbate the effects of adverse conditions, including extreme or variable temperatures, on their development [19,20]. Altricial nestlings often hatch asynchronously, with some nestlings hatching a day or two later than their brood mates as a result of incubation beginning before all eggs are laid [21]. Hatching asynchrony typically results in size asymmetry, with younger nestlings remaining smaller than their brood mates across development [21]. This size asymmetry may affect access to resources—older nestlings outcompete the younger nestlings for food, and younger nestlings often grow more slowly and even die from starvation [21–23]. Heterogeneity in access to resources within a brood may then drive differences in susceptibility to adverse conditions such as temperature. Additionally, body size is a

critical determinant of thermoregulatory ability [6,24], but most studies have investigated the relationship between size and thermoregulation among rather than within species [25,26]. As such, investigating whether the size of a nestling relative to its brood mates influences responses to temperature can help elucidate how early-life social experience shapes nestling responses to environmental challenges [22,27–29].

Parental care, including food provisioning, is another aspect of the social environment that may shape nestling responses to adverse conditions [19,20,30]. Nestlings are entirely dependent on food provisioned by parents to provide energy for growth and thermoregulation. As such, nestlings that receive more parental care may be less susceptible to the negative effects of temperature and other adverse conditions because they have more energetic reserves for thermoregulation and growth. Indeed, research on burying beetles (*Nicrophorus vespilloides*), an invertebrate that displays elaborate biparental care like many altricial bird species, indicates that higher levels of parental care can buffer young from effects of cold temperatures [31]. Similarly, studies of cooperatively breeding bird species suggest that the presence of helpers—additional, non-parent individuals that care for offspring (an example of alloparental care)—may mitigate negative impacts of low rainfall on nestlings [19] (but see [30]). Finally, adverse weather conditions may lead to reduced food provisioning. For instance, provisioning rates decrease in hot or cold temperatures due to reduced availability of insect prey [11,32] and parents diverting more time and energy to their own thermoregulation [33,34]. Similarly, food availability and provisioning rates often decrease during periods of precipitation or high wind, especially for aerial insectivores [29,35–38]. Therefore, it is plausible that parental feeding influences nestling vulnerability to adverse conditions, including extreme temperatures, in altricial birds.

In wild barn swallows (*Hirundo rustica erythrogaster*), we asked three questions and explored related hypotheses about the effects of near-nest temperature on nestling growth. First, we asked: Does the effect of temperature on nestling mass differ when the exposure is assessed during early versus late development? Given that temperature exposures during different stages of development differentially affect nestling metabolic rate and thermoregulatory strategies (e.g., [5,6,15]), we hypothesized that, due to differences in metabolic costs, exposure to extreme and variable temperatures early in development would have a stronger effect on nestling mass than exposure later in development. We predicted that hot, cold, and variable temperatures from hatch to day five (early in development), before expected development of thermoregulation, would have stronger negative effects on nestling mass before fledging than temperatures from day six to twelve (late in development) (Fig 1).

Our second and third questions asked whether social experiences modify the effects of temperature on nestling growth. For question two, we asked: Does the effect of temperature on nestling mass depend on whether a nestling is the smallest in the brood? For question three, we asked: Does the effect of temperature on nestling mass depend on the amount of feeding provided by parents? Differences in nestling size are associated with heterogeneity in access to resources [21–23] and can impact thermoregulatory ability [6,24]. In addition, previous studies report that parental care can buffer nestlings from the effects of adverse conditions [19,31]. Therefore, we hypothesized that disadvantageous social environments (e.g., being the smallest in the brood, lower levels of parental feeding) would exacerbate the negative impacts of extreme and variable temperatures on nestling mass due to reduced access to resources. We predicted that hot, cold, and variable temperatures would have stronger effects on nestling mass for the smallest nestling in the brood relative to other brood mates and that temperature exposure would have stronger effects on nestling mass at low than high levels of parental feeding (Fig 1). We set out to test these hypotheses in an exploratory manner, as a step toward understanding how the developmental and social conditions under study may shape temperature effects on nestlings.

## Methods

### Study system

We monitored nesting attempts from wild barn swallows during the summer of 2021 at seven breeding sites in Boulder County, CO (approximate latitude: 40.070733, longitude: −105.232863, altitude: 1,582 meters). These sites are monitored

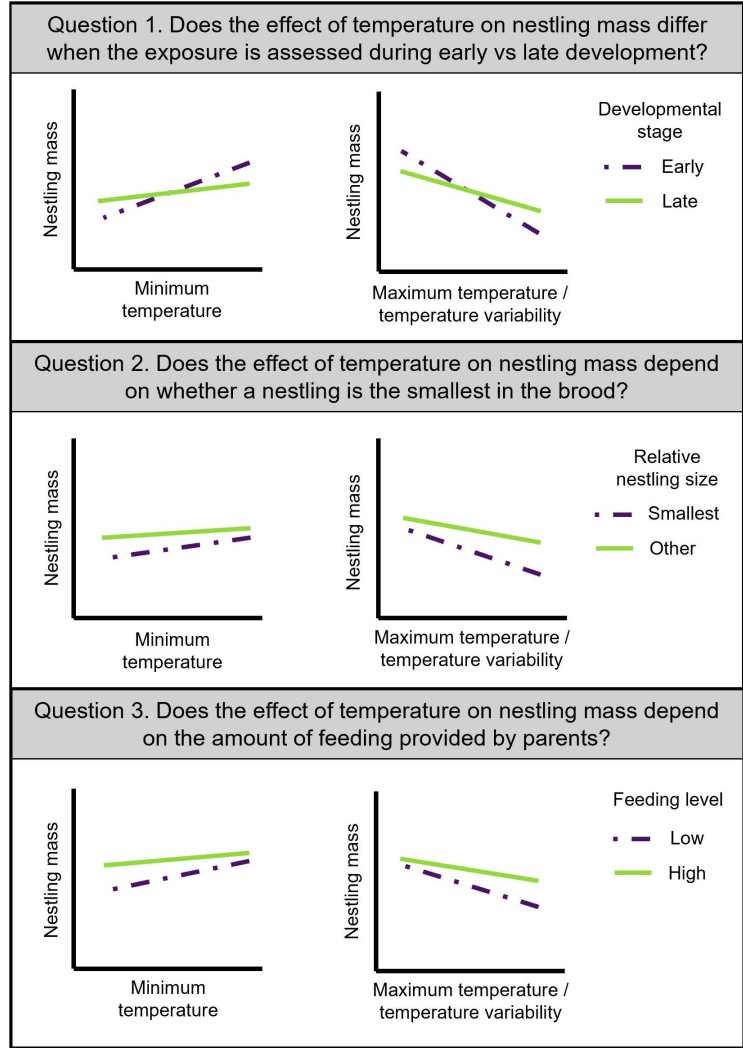

**Fig 1. Research questions and predicted results for our hypotheses about temperature effects on wild barn swallow nestling mass before fledging.** Feeding level refers to the average nest-level feeding rate assessed at three points in time across nestling development.

as part of a long-term study of barn swallows and are occupied by colonies ranging in size from one to 50 breeding pairs. Barn swallows build open mud cup nests and typically lay clutches of three to five eggs, which hatch asynchronously over approximately one to two days [39]. Hatching asynchrony results in a size hierarchy that persists over the course of development [40]. Offspring sex is associated with neither hatch order nor external morphological differences in this species [41]. Both parents in a social pair provision food until nestlings fledge in this socially monogamous species [39]. Fledging typically occurs around 21 days post-hatch (range 15 to 27 days) in North American populations [39].

## Study design overview

We collected environmental, demographic, and behavioral data at 31 first-brood nests from May through July 2021, and we measured the morphometric and physiological traits for 113 nestlings across development (Fig 2). Nests were checked twice per week to track nestling phenology and fate—typically, we visited a given nest every three to four

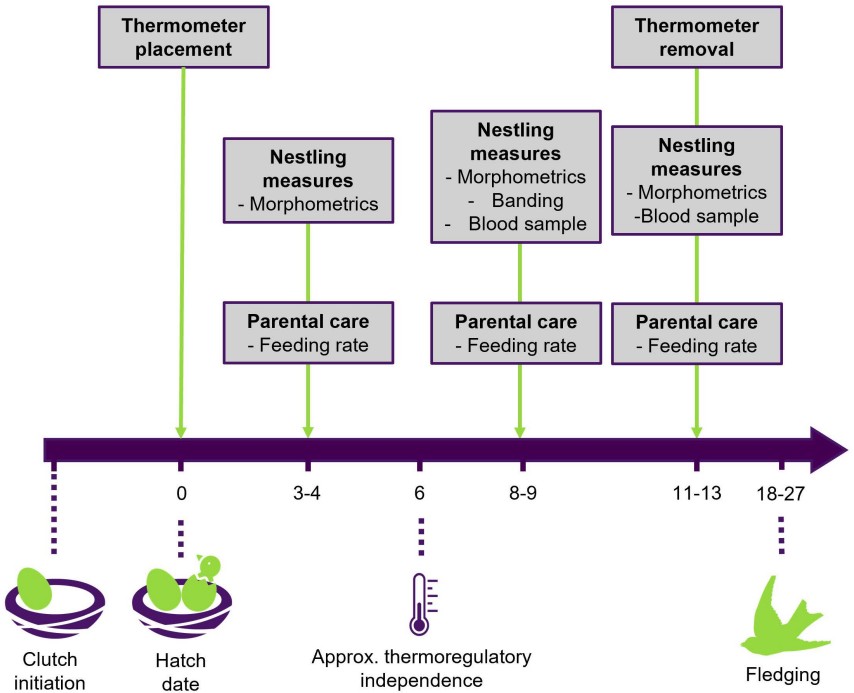

**Fig 2. An overview of the study design indicating the types of data collected across the nestling period at three time points in wild barn swallows in Boulder County, CO.**

days, and the time between nest checks did not exceed five days. When nests were close to their estimated hatch date (based on clutch initiation date), they were checked every two days. During the first check after nestlings hatched, we estimated their ages (hatch day = day zero) based on feather emergence and other reliable developmental characteristics, such as wetness (on hatch day) and ability to raise their head [42]. We took morphological measurements of nestlings and conducted one-hour observations of parental care behavior at three time points: days 3–4, days 8–9, and days 11–13. We monitored temperature using loggers placed outside the nest, 10–28 cm from the nest edge, from hatch through last nestling measures (days 11–13).

## Nestling measures

We collected standard morphometric measurements for each nestling at three time points ([43]; Fig 2). We measured mass to the nearest 100th of a gram using a digital scale and right-wing length to the nearest half millimeter as the distance from the carpal joint to the distal end of the second phalanx or primary wing feather, whichever was longer. On days 8–9, we placed a USGS metal band on the tibiotarsus of each nestling, allowing us to track individuals through development. We also collected two fasting blood samples from each nestling at each of two time periods—on days 8–9 and 11–13—as part of a separate study [43]. Each blood sample consisted of several drops of blood (<30 μL), collected via venipuncture of the brachial vein [43].

Using days 8–9 measures (the earliest age at which we could begin tracking individuals), we created a categorical variable of relative nestling size, where the smallest nestling in each nest based on right-wing length was classified as 'min' (reference group), and the remaining nestlings were classified as 'other'. We choose to compare the smallest nestling to all other nestlings because, in North American barn swallows, incubation generally begins when the penultimate

egg is laid [39]; As a result, it is common for one nestling to hatch approximately one day later than its brood mates, and we expected the largest size disparity to be between this last hatched nestling and all others [40]. Our classification was based on right wing length, rather than mass, because wing length is not influenced by short-term fluctuations like mass [44–46]. Because we were unable to track individuals until days 8–9, we could not determine the exact age of individual nestlings. Therefore, we are unable to tease apart the effects of age and mass in the relationship between relative nestling size and vulnerability to temperature exposure.

**Parental care observations**

We conducted one-hour focal observations of each nest at three time points (see details in [43]; Fig 2). Some nests are missing data for one or more time points due to logistical issues. The start of each observation was preceded by the collection of morphometric measurements and blood samples from nestlings (see "Nestling Measures"). After these measures, observers watched the nest from a blind, and the birds were given about 15 minutes to habituate to our presence before the observation commenced (following [43,47]). Observations began between 5:45 and 8:00 am, and their duration ranged from 50.00 to 62.58 minutes, with a mean of 59.70. We did not complete observations during rain or high wind conditions (observation mean wind speed = 0.15, range 0 to 4.02 m/s).

When an observer could not be present, we instead recorded nests using GoPro cameras, and videos were scored by the same observers who conducted the in-person observations. For these camera observations, we set up camera mounts the day before recording to allow birds to habituate, and we waited for about 15 minutes after the cameras were started to begin scoring behavior. Parental care behaviors are consistent when scored by different observers and between in-person and camera observations [47].

During each observation, the observer logged parental care behavior in real-time on iPads or iPhones (Apple Inc.) using "Animal Behavior Pro" [48]. For this study, we focused on feeding visits, where the parent delivers food to a nestling. However, we recorded additional behaviors (see full ethogram in S1 Table) during each observation period. Using these data, we examined the correlation of feeding rate and the proportion of time spent brooding using Spearman rank correlation because both behaviors may influence nestling response to temperature. Given biparental care in this species, we recorded the total number of behaviors for both social parents.

We created an index of the level of parental feeding by summarizing feeding rates from both parents across all stages of nestling development. Specifically, we used a generalized linear mixed effects model, in which total feeding count by both parents measured at each developmental stage was the outcome and nest ID was a random intercept. To fit models, we used the 'lme4' package, version 1.1.35.5 [49] in R version 4.4.1 [50]. We compared models with several error distributions and link functions using diagnostic plots and tests from the package 'DHARMa,' version 0.4.6 [51]. We selected a negative binomial distribution and log link for our final model because this appropriately handled overdispersion in the count data. The model included an offset for total observation time and several covariates that could influence parental feeding behaviors: number of days since the first nestling hatched, number of nestlings in the nest, the median temperature near the nest during the observation period, and the duration of time between removing nestlings from the nest for morphometric and physiological measures and the start of the observation. From this model, we extracted the best linear unbiased predictors (BLUPs) (following [43]). Because parental feeding is an effect modifier in our models, we created a two-level categorical variable for stratified analyses by classifying BLUPs as 'low' (lowest half) or 'high' (highest half) parental feeding (S1 Fig).

Using a mixed effects model in this way allowed us to summarize parental care across developmental stages while controlling for environmental conditions and nest-level variables that may influence parental care. Moreover, this method is an efficient use of our data because it allows for missing data via shrinkage of nest-level estimates toward the global sample mean. The approach of carrying BLUPs forward from one model to another has been criticized due to its failure to propagate uncertainty [52]. However, because summarizing parental feeding using BLUPs should not substantively shift

nests from one parental feeding level to another, this approach is justified in our case, given the advantages described above.

## Near-nest temperatures

We hung thermometers (Govee) in small mesh bags adjacent (10–28 cm) to each open cup nest and logged near-nest temperature and humidity every 15 minutes throughout the data collection period (Fig 2). At one nest, we are missing two days of temperature measurements at the end of the nestling period. We calculated the minimum temperature, maximum temperature, and temperature variability (defined as the interquartile range) at each nest from day zero (hatch) to the end of day five (early development) and from day six until the last measure of nestling mass on days 11–13 (late development). The timing of these temperature summaries correspond with before and after the putative development of thermoregulatory independence, respectively. We chose these time periods based on previous research in swallows and other altricial birds [15,53–55], which found that nestling swallows begin to develop the ability to thermoregulate independently around four to five days post-hatch, and the age of the effective homeothermy (defined here as ability to maintain relatively constant body temperature under natural conditions—in the nest with brood mates) is approximately six days of age. Finally, we calculated the minimum temperature, maximum temperature, and temperature variability at each nest over the entire nestling period, from hatch day to the last measure of nestling mass.

## Ethics statement

All research on wild barn swallows was approved by the Institutional Animal Care and Use Committee protocol (permit no. 1303.02), and all research was performed in accordance with relevant guidelines and regulations. In addition, this work was conducted under the Bird Banding Lab, master permit bander ID 23505. Capture using mist nests in Boulder, Weld, and Jefferson Counties, Colorado, was approved by the Colorado Parks and Wildlife, issued under permit license number 21TRb2005. As part of our data collection procedure, no birds were sacrificed or sedated with anesthesia. When removing nestlings from their nests or handling adult birds captured in mist nets, we followed standard protocols to minimize handling time and stress. For nestlings, these protocols included minimizing their time out of the nest, containment of all nestlings together in small cloth-lined containers, and providing a mild heat source when extraction occurred on cool mornings. For adults, birds were immediately removed from mist nets, placed in opaque cloth bags to minimize stress, and handled and measured by trained personnel. Clean cloths / bags were used for each brood of nestlings and each adult bird. After measurements were taken, nestlings were immediately returned to their nests, and adults were released at the location where they were captured. Some swallow nests were located on private property and were monitored with permission from the landowners.

## Statistical analysis

We conducted three sets of formal analyses aligning with our questions. Across all questions, we investigated as explanatory variables: 1) minimum temperature, which captures exposure to extreme cold, 2) maximum temperature, which captures exposure to extreme heat, and 3) temperature variability, which captures exposure to fluctuations between cold and hot [9,10]. We focused our reporting and interpretation of results on estimates of effect sizes and 95% confidence intervals, providing information on the strength and uncertainty of our effect estimates [56], as well as p-values interpreted using the language of evidence [57]. P-values less than or equal to 0.05 were interpreted as providing moderate to strong evidence for an effect, p-values between 0.05 and 0.10 as providing marginal evidence, and p-values greater than 0.10 as providing little to no evidence [57]. To assess goodness of fit, we estimated marginal $R^2$ (variance explained by fixed effects alone) and conditional $R^2$ (variance explained by full model, including fixed and random effects) for each model [58], using the package 'MuMIn', version 1.48.4 [59].

Across all analyses, we report both unadjusted associations and adjusted associations from multiple variable regression models. Examining both unadjusted and adjusted associations allowed us to see if the estimate of interest changes after controlling for other variables. For adjusted models, we included hatch date and number of nestlings in a nest as covariates. For all models, we z-score standardized (subtracted the mean and divided by the standard deviation) numeric explanatory variables and covariates to aid in model fit and comparison of effect sizes. All models included a random intercept for nest ID to account for non-independence of measures taken at the same nest. We assessed inclusion of a random effect for site ID to account for non-independence of measures taken at the same site, but this effect was removed to avoid model overfitting because it explained very little or zero variance across models, and its inclusion resulted in a singular fit for some models. Removal of site ID from models in which it explained a little variance did not substantially affect the direction, magnitude, precision, or significance of effects.

Model assumptions were checked using diagnostic plots. Specifically, we checked whether each model met the assumptions of a Gaussian error distribution with homogenous variance by examining plots of residuals versus fitted values, a histogram of the residuals, a normal quantile-quantile plot, and Cook's distance and standardized residual influence plots, using the package 'car', version 3.1.3 [60]. For some models, we detected influential outliers, which we defined as having a Cook's distance greater than one or a standardized residual value greater than three. When influential outliers were present, we conducted additional analyses, where we ran the model with and without the outlier(s) and compared the results.

For question one, to assess the effects of temperature on nestling mass at two different developmental periods (early and late development), we separately ran two sets of linear mixed models for each of our three temperature variables (minimum, maximum, and variability) (Fig 3): 1) In the first set of three models, each temperature variable collected in early development (from hatch to the end of day five) was included as the explanatory variable of interest and nestling mass on days 11–13 was the outcome; 2) the second set of models was the same except that temperature variables were quantified in late development (from day six to last nestling measures on days 11–13). To aid our inferences, we examined the correlation of each of our temperature variables between early and late development using Spearman rank correlation. We conducted additional analyses to assess the robustness of our results to differences in the selected cut-off for development of thermoregulatory independence. Specifically, we ran the same set of models described above, except using cut-off ages of five days post-hatch and seven days post-hatch, rather than six days post-hatch, for the development of thermoregulatory independence.

For questions two and three, we tested for effect modification of temperature effects on nestling mass by relative nestling size and level of parental feeding, respectively (Fig 3). Effect modification refers to a situation in which the effect of X on Y depends on a third variable, the effect modifier [61]. In our analysis, relative nestling size (question two) and level of parental feeding (question three) were treated as effect modifiers. To determine whether effect modification was present, we ran analyses stratified by levels of relative nestling size or parental feeding. Results from stratified models provide insight into the magnitude and direction of the effects of temperature on nestling mass for different relative nestling sizes or levels of parental feeding. In addition, for each question, we ran a model with unstratified data including an interaction term between temperature and the effect modifier (relative nestling size or parental feeding). The estimate, confidence interval, and p-value for the interaction term were used to inform the interpretation of stratified results. We present stratified results even in the absence of significant interaction terms, given that interaction terms can be nonsignificant when effect modification is present due to factors such as low power, the scale of measurement selected, and the presence of non-linear relationships [62–65].

For question two, we modeled the effect of each temperature variable measured across the nestling period on nestling mass on days 11–13, stratified by nestling size: 'min' = the smallest nestling in the nest and 'other' = all other nestlings. For question three, we modeled the effect of temperature measured across the nestling period on nestling mass on days 11–13, stratified by parental feeding, which included two levels: 'low' = parental feeding below the 50th percentile and 'high' = parental feeding above the 50th percentile. We conducted additional analyses to assess the robustness of our results to differences

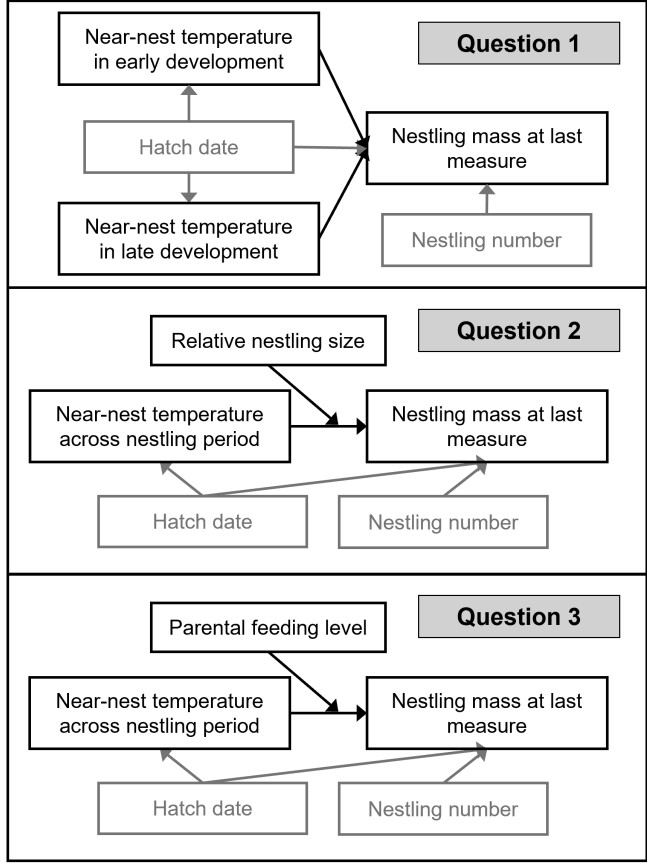

**Fig 3. Conceptual graphs displaying hypothesized causal relationships investigated in each of our three questions.** The explanatory and response variables have black boxes, while confounders and precision covariates have gray boxes. Arrows represent hypothesized relationships. An arrow pointing directly into the middle of another arrow represents effect modification.

in the selected number of strata for parental feeding. Specifically, we ran the same set of models described above, except using three strata for parental feeding: 'low'=parental feeding below the 33rd percentile, 'med'=parental feeding between the 33rd and 67th percentile, and 'high'=parental feeding above the 67th percentile, instead of two strata.

All statistical analyses were carried out using R version 4.4.1 [50]. The 'tidyverse' package, version 2.0.0 [66] was used for data cleaning and organization. The 'lme4' package, version 1.1.35.5 [49] was used to fit models. Bootstrapped confidence intervals were obtained for model coefficients using the package 'boot', version 1.3.30 [67,68]. P-values for two-tailed t-tests for each coefficient were obtained using Satterthwaite degrees of freedom method, via the package 'lmerTest' version 3.1.3 [69]. Results were visualized using the package 'ggplot2' version 3.5.1 [70], 'ggeffects' version 1.7.2 [71], and 'ggpubr' version 0.6.0 [72].

## Results

### Background characteristics

All results for background characteristics are presented as mean±SD, unless otherwise stated. Of the 113 nestlings in this study, 108 survived until days eight to nine, and 106 survived until days 11−13. Clutch size was 4.31±0.74 eggs. Brood size was 3.42±0.96 nestlings, and hatch date was 20.58±10.05 days since June 1 (S2 Table). Eggs hatched

asynchronously over a period of up to two days—the mean difference in age between the first and last hatched nestling was 1.03 days (range = 0 to 2 days). For the near-nest temperature measurements across the nestling period, the mean temperature was 23.99 ± 1.32 **°C**, the minimum temperature was 13.10 ± 1.80 °C, the maximum temperature was 36.90 ± 2.96 °C, and the temperature variability was 9.26 ± 2.43 °C (S2 Table; S2 Fig)**.** On days 8−9, right wing length was 26.8 ± 5.55 mm for the smallest ('min') nestlings and 32.4 ± 4.26 mm for 'other' nestlings, and mass was 11.60 ± 2.61 g for the smallest ('min') nestlings and 13.60 ± 2.30 g for 'other' nestlings. On days 11−13, the timepoint at which the outcome was measured, wing length was 49.20 ± 8.09 mm for the smallest ('min') nestlings and 54.40 ± 4.76 mm for 'other' nestlings, and mass was 17.20 ± 2.75 g for the smallest ('min') nestlings and 17.70 ± 1.91 g for 'other' nestlings. For measures of parental care, the total feeding rate (counts/hour) was 12.21 ± 6.92 on days 3−4, 12.73 ± 8.08 on days 8−9, and 21.62 ± 12.94 on days 11−13 (S2 Table). Feeding rate and proportion of time spent brooding during the observation were negatively correlated when nestlings were 8−9 and 11−13 days old (days 8−9: n = 26, rho = −0.58, p = 0.002; days 11−13: n = 25, rho = −0.43, p = 0.03), but not when they were 3−4 days old (n = 25, rho = −0.25, p = 0.22).

### Question 1. Does the effect of temperature on nestling mass differ when the exposure is assessed during early versus late development?

Higher minimum temperatures in early development (β = 1.16 g per 1 SD °C [95% CI: 0.32, 2.00], p = 0.01) but not late development (β = 0.29 g per 1 SD °C [95% CI: −0.45, 1.06], p = 0.46) were associated with higher nestling mass at last measure (S3 Table; Fig 4A). There was marginal evidence that higher maximum temperatures in early development were associated with lower nestling mass (β = −0.66 g per 1 SD °C [95% CI: −1.32, −0.04, p = 0.053; S3 Table; Fig 4B). Similarly, higher maximum temperatures in late development were associated with lower nestling mass (β = −1.13 g per 1 SD °C [95% CI: −1.76, −0.52], p = 0.001 S3 Table; Fig 4B). Greater temperature variability in early (β = −1.41 g per 1 SD °C [95% CI: −2.07, −0.77], p = 0.0003) and late (β = −1.33 g per 1 SD °C [95% CI: −1.95, −0.70], p = 0.0003) development were associated with lower nestling mass (S3 Table; Fig 4C).

   The magnitude, direction, and precision (width of CI), and significance of estimates of interest were similar when models were run using five days or seven days post-hatch, rather than six days post-hatch, as the cut-off for age for early versus late development, and when influential outliers were excluded from analyses**.** For minimum temperature, temperatures in early development were not significantly correlated with temperatures in late development (n = 31, rho = 0.26, p = 0.15). On the other hand, there was a significant positive correlation between temperatures in early and late development for maximum temperature (n = 31, rho = 0.57, p = 0.0007) and temperature variability (n = 31, rho = 0.81, p < 0.0001).

### Question 2. Does the effect of temperature on nestling mass depend on whether a nestling is the smallest in the brood?

In stratified analyses, higher minimum temperatures across the nestling period corresponded with higher nestling mass for both the smallest nestlings in the nest (β = 1.75 g per 1 SD °C [95% CI: 0.43, 3.14], p = 0.02; S4 Table; Fig 5A) and for other nestlings (β = 1.30 g per 1 SD °C [95% CI: 0.40, 2.31], p = 0.01; S4 Table; Fig 5A). In unstratified analyses, there was marginal evidence for an interaction between minimum temperature and relative nestling size (β = −0.36 g per 1 SD °C [95% CI: −0.77, 0.03], p = 0.08).

   Similarly, higher maximum temperatures and greater temperature variability across the nestling period were associated with lower nestling mass for the smallest nestlings in the nest (maximum: β = −1.42 g per 1 SD °C [95% CI: −2.48, −0.40], p = 0.01; variability: β = −1.95 g per 1 SD °C [95% CI: −2.98, −0.92], p = 0.001; S4 Table; Fig 5B, 5C) and for other nestlings (maximum: β = −0.85 g per 1 SD °C [95% CI: −1.47, −0.27], p = 0.01; variability: β = − 1.50 g per 1 SD °C [95% CI: −2.15, −0.89], p = 0.0001; S4 Table; Fig 5B, 5C). There was marginal evidence for an interaction between maximum temperature and relative nestling size (β = 0.37 g per 1 SD °C [95% CI: −0.08, 0.79], p = 0.096) and between temperature variability and relative nestling size (β = 0.39 g per 1 SD °C [95% CI: −0.03, 0.81], p = 0.07).

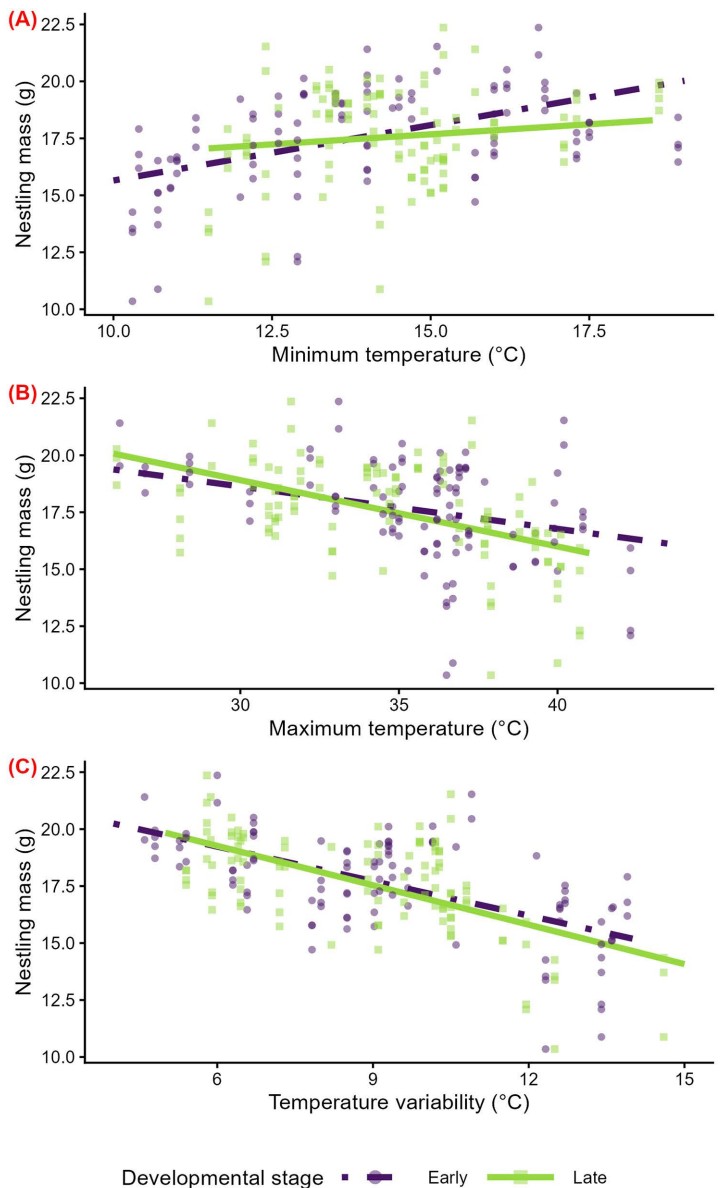

**Fig 4. Temperature effects in early and late development on the mass of 11–13-day-old nestlings.** Predicted relationships of minimum temperature (A), maximum temperature (B), and temperature variability (C) and nestling mass from separate linear mixed models for each developmental stage—in early ('Early', n = 106) development, from hatch through day five, and late ('Late', n = 106) development, from day six to final measures at 11-13 days post-hatch. Model predictions are displayed as lines, and raw data are displayed as points. Colors, line types, and shapes correspond to the developmental stage.

## Question 3. Does the effect of temperature on nestling mass depend on the overall amount of feeding provided by parents?

In stratified analyses with two parental feeding levels, higher minimum temperatures across the nestling period were associated with higher nestling mass among nests that received low parental feeding (β = 2.18 g per 1 SD °C [95% CI: 0.48, 4.01], p = 0.03; S5 Table; Fig 6A). On the other hand, there was little evidence that minimum temperatures were

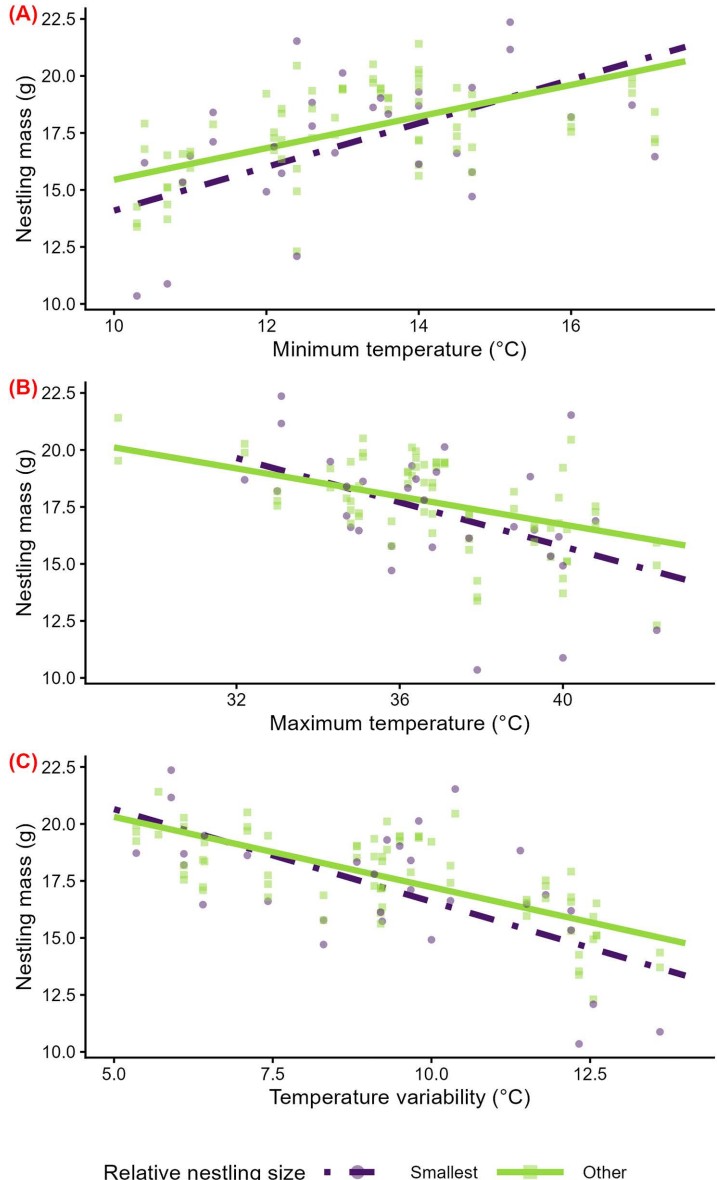

**Fig 5. Temperature effects on the mass of 11–13-day-old nestlings that are the smallest in their brood versus other nestlings.** Predicted relationships of minimum temperature (A), maximum temperature (B), and temperature variability (interquartile range) (C) across the nestling period and nestling mass from linear mixed models, stratified by relative nestling size—the smallest nestling ('Smallest', n = 31) and all other nestlings ('Other', n = 72)—at days 8–9 post-hatch. Model predictions are displayed as lines, and raw data are displayed as points. Colors, line types, and shapes correspond to the relative nestling size. Because relative nestling size was determined at days 8–9 and some nestlings did not survive to final measures on days 11–13 (n = 2 nestlings that died between days 8–9 and 11–13), only 'Smallest' or 'Other' nestling points are present for some nests.

associated with nestling mass among nests that received high parental feeding ($\beta$ = 0.97 g per 1 SD °C [95% CI: −0.06, 2.12], p = 0.102; S5 Table; Fig 6A). In unstratified analyses, there was evidence for an interaction between minimum temperature and parental feeding ($\beta$ = −1.37 g per 1 SD °C [95% CI: −2.61, −0.19], p = 0.03).

In stratified analyses, maximum temperatures across the nestling period were not associated with nestling mass among nests that received low parental feeding ($\beta$ = −1.04 g per 1 SD °C [95% CI: −2.22, 0.12], p = 0.11; S5 Table;

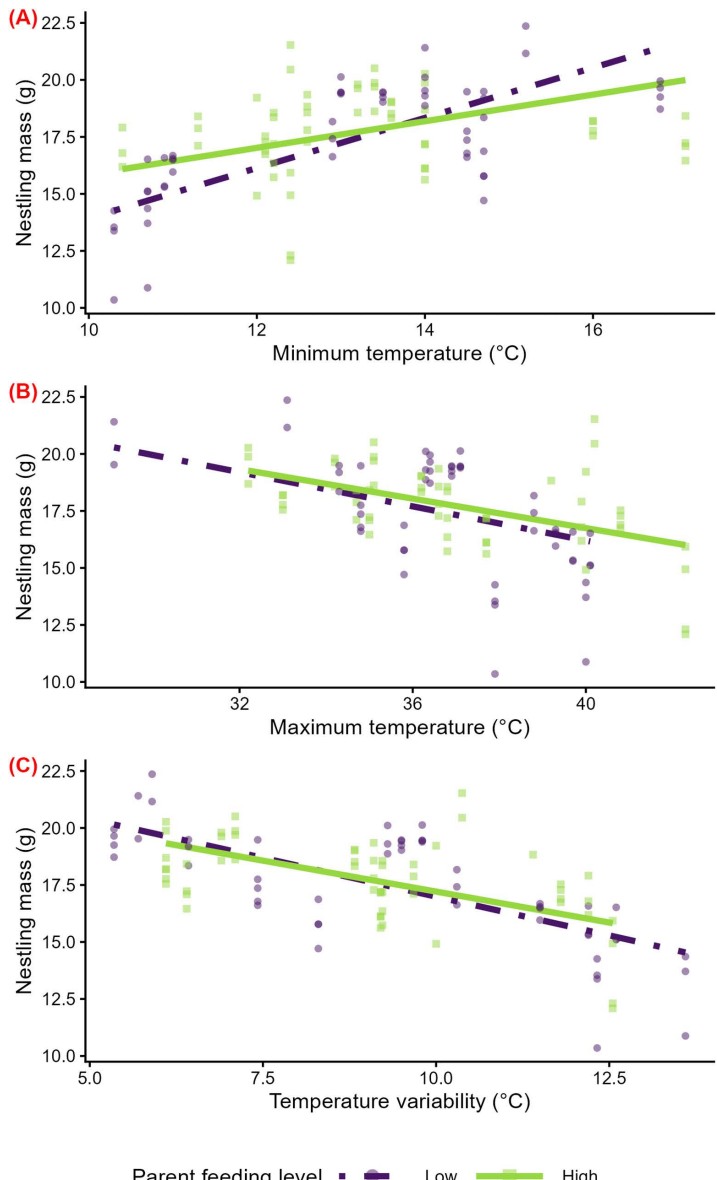

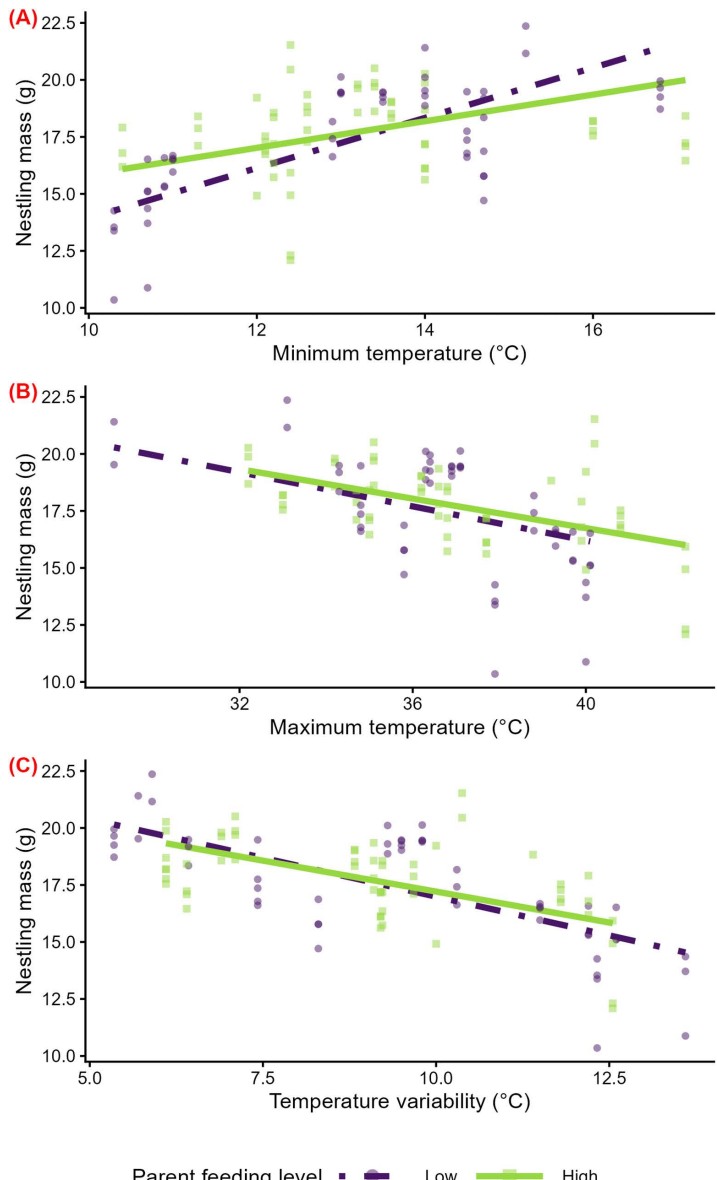

**Fig 6. Temperature effects on the mass of 11–13-day-old nestlings at nests receiving low or high parental feeding.** Predicted relationships of minimum temperature (A), maximum temperature (B), and temperature variability (interquartile range) (C) across the nestling period and nestling mass from linear mixed models, stratified by two levels of parent feeding—low ('Low', n=51) and high ('High', n=55). Model predictions are displayed as lines, and raw data are displayed as points. Colors, line types, and shapes correspond to the level of parent feeding.

Fig 6B). In contrast, higher maximum temperatures were associated with lower nestling mass among nests that received high parental feeding (β=−0.93 g per 1 SD °C [95% CI: −1.59, −0.33], p=0.01; S5 Table; Fig 6B). Higher temperature variability was associated with lower nestling mass among nests that received both low parental feeding (β=−1.82 g per 1 SD °C [95% CI: −3.15, −0.53], p=0.02; S5 Table; Fig 6C) and high parental feeding (β=−1.14 g per 1 SD °C [95% CI: −1.79, −0.59], p=0.004; S5 Table; Fig 6C). In unstratified analyses, there was no evidence for an interaction between maximum temperature and parental feeding level (β=0.52 g per 1 SD °C [95% CI: −0.69, 1.77], p=0.41) or between

temperature variability and parental feeding level ($\beta$ = 0.80 g per 1 SD °C [95% CI: −0.21, 1.88], p = 0.15). The magnitude, direction, precision (width of CI), and significance of estimates of interest were similar when influential outliers were excluded from analyses. Additionally, in the analyses with three parental feeding strata, rather than two, the qualitative patterns in stratified analyses, and the magnitude, direction, and precision, and significance of the interaction term in unstratified analyses, were similar (S6 Table; S3 Fig).

## Discussion

We found mixed support for our hypothesis that exposure to extreme and variable temperatures early in development have a stronger effect on nestling mass than exposure later in development, which may be due to differences in metabolic costs. Consistent with our hypothesis, we found that higher minimum temperatures in early but not late development--*before but not after* the putative development of thermoregulatory independence at six days post-hatch—were associated with higher nestling mass (Fig 4A). Contrary to our hypothesis, higher maximum temperatures and greater temperature variability in both early and late development—*before and after* putative development of thermoregulatory independence—were associated with lower nestling mass (Fig 4B and 4C).

Similarly, some of our results but not others supported our hypothesis that disadvantageous social environments exacerbate the negative impacts of extreme and variable temperatures on nestling mass, possibly due to the nestlings' reduced access to resources. Consistent with our hypothesis, we found marginal evidence that minimum temperature, maximum temperature, and temperature variability have stronger effects on the mass of the smallest nestling than other brood mates (Fig 5), as indicated by a marginally significant statistical interactions of temperature with relative nestling size. Additionally, in support of our hypothesis, minimum temperature had stronger effects on nestling mass in nests receiving low than high levels of parent feeding (Fig 6A), as evidenced by a significant statistical interaction of parental feeding level and minimum temperature. On the other hand, contrasting with our hypothesis, there was no evidence for quantitative differences in the effects of maximum temperature and temperature variability on nestling mass across levels of parental feeding (Fig 6C and 6D).

### Minimum temperature, maximum temperature, and temperature variability have differential effects on nestling mass

Our collective set of results suggests that exposure to lower minimum temperatures, higher maximum temperatures, and greater temperature variability were associated with lower nestling mass, consistent with literature reporting that exposure to cold [11,73,74], hot [75–77], and variable temperatures [78] may all lead to reduced nestling growth. However, in contrast to our results, some studies report no effect of extreme or variable temperatures on nestling growth [79–81]. Ecological factors, such as habitat, diet, and nest type, may shape susceptibility to negative effects of temperature across populations and species [82]. For example, insectivorous species may be more sensitive to extreme temperatures than granivorous species because insect availability is more strongly influenced by weather conditions, including rain, wind, speed, and temperature, than seed availability [35,83,84]. Additionally, open-cup nesting species may be more sensitive to extreme temperatures than cavity-nesting species because open-cup nests may be less thermally buffered than cavity nests [82]. Given that barn swallows are insectivorous and build open cup nests, it is perhaps unsurprising that nestling mass is negatively affected by cold, hot, and variable temperatures in this species.

### Effects of minimum temperature on nestling mass may differ across nestling ontogeny

There are several possible explanations for our finding that lower minimum temperatures in early but not late development were associated with lower nestling mass. Nestlings may be especially sensitive to cold temperatures before thermoregulatory independence develops (approximately halfway through the nestling period [6]) because colder ambient temperatures result in colder body temperatures at times when the parents are away from the nest, and colder temperatures slow

metabolic processes in young nestlings [85]. In older nestlings, improved thermoregulation allows for a more constant body temperature, and metabolic rate increases when ambient temperatures are outside the thermal neutral zone [6]. Additionally, parents must brood young nestlings frequently, especially during cold temperatures, and they are generally unable to feed and brood nestlings simultaneously. This trade-off between brooding and feeding may lead to larger effects of cold temperatures on the growth of young than old nestlings. Third, due to their higher surface area to volume ratio and lack of feathers, young nestlings may dissipate heat more quickly than older nestlings, which may be a disadvantage under cold conditions [6,8,81]. Finally, changes in specific thermoregulatory behaviors (e.g., panting or shivering, [86,87]) and mouth surface area, which may affect evaporative cooling via panting, across ontogeny might also contribute to differences in temperature effects on young and old nestlings.

Our finding that higher maximum temperatures and greater temperature variability in both early and late development were associated with lower nestling mass suggests that nestlings may be sensitive to heat and environmental instability regardless of thermoregulatory status. However, the mechanisms underlying this temperature sensitivity may shift across ontogeny. Early in development, nestlings may be susceptible to negative impacts of extreme heat because they may gain heat more quickly than older nestlings due to their high surface area to volume ratio and lack of feathers [6,8,81]. Late in development, after development of thermoregulatory independence, nestlings may be sensitive to hot temperatures because evaporative cooling is energetically costly and may quickly lead to dehydration [77,88]. In addition, nestlings may have limited ability to cool themselves [88], which may be exacerbated in older nestlings by reduced heat dissipation due to lower surface area to volume ratio, feather development, and crowding in the nest (though reduced heat transfer may be an advantage when ambient temperatures exceed body temperature). Finally, the similar patterns we observed across time periods may have been influenced by positive correlations between temperatures in early and late development for maximum temperature and temperature variability. These correlations suggest that nests may experience similar temperature conditions in early and late development, and our findings may reflect the consistent effects of these environmental conditions across development on mass, rather than the separate effects of conditions during each developmental stage. Taken together, these developmental-stage-specific mechanisms and similar temperature conditions across development may explain why hot and variable temperatures negatively impacted nestlings in both early and late development.

### Effects of temperature on nestling mass may differ for the smallest nestling compared to other brood mates

Our findings suggest that, although nestlings seem to experience negative effects of extreme and variable temperatures regardless of their relative size, the magnitude of these effects may be greater for the smallest nestling in the nest than for other nestlings, as indicated by the marginally significant temperature by size interaction terms. There are several possible explanations for why the magnitude of temperature effects on nestlings may differ according to their relative size within a nest. First, animals with a smaller body size are more vulnerable to extreme temperatures because they are less able to regulate their body temperature—they dissipate heat more quickly when ambient temperature is below body temperature, and they absorb heat more quickly when ambient temperature exceeds body temperature [6,24]. This explanation involving thermoregulatory constraint aligns with studies of other species finding that extreme temperatures have stronger effects on the survival [27] and growth [22,29] of the smallest or last hatched nestling than other brood mates.

Second, smaller nestlings may face a competitive disadvantage compared to their larger brood mates. Larger nestlings often outcompete younger nestlings for resources, which are crucial for thermoregulation and other developmental processes like growth [21,22]. For example, in a study of blue tits (*Cyanistes caeruleus*), higher levels of rainfall had stronger positive impacts on the mass of early hatched nestlings than that of later hatched nestlings, which the authors suggested may be due to the greater ability of early hatched nestlings to secure food from parents than late hatched nestlings [28]. Additionally, competitive dynamics may shift under adverse conditions—extreme temperatures may exacerbate the competitive disadvantage of small nestlings [89]. Therefore, we might expect that larger nestlings that more efficiently thermoregulate or that can acquire more resources are likely to be less susceptible to extreme temperatures.

### Effects of minimum temperature on nestling mass may differ by level of parental feeding

Our finding that cold temperatures may have stronger negative effects on nestling mass for the nests receiving low versus high levels of feeding is consistent with several studies in non-avian species and cooperatively breeding birds, which found that higher levels of parental care reduced the negative impacts of adverse weather conditions (cold temperatures and periods of low rainfall, respectively) on offspring [19,31] (but see [30]). On the other hand, our finding that the effects of hot and variable temperatures on nestlings do not depend on level of parental feeding contrasts with these previous studies. In combination, our findings may suggest that parents can buffer nestlings from cold temperatures by shifting their overall feeding level, but their ability to buffer nestlings from hot and variable temperatures may be more limited.

While we and others have focused on feeding, feeding may co-vary with other parental care behaviors, such as brooding, that also influence nestlings' responses to temperature exposure. If parental feeding and brooding positively covary, as some studies have reported [90,91], then any modification of temperature effects on nestling mass by parental care may be driven by active parental regulation of nest temperatures, in addition to or instead of access to higher energetic resources via food provisioning. However, in our study population and other studies: [92,93], feeding and brooding are negatively correlated (at 8–9 and 11–13, but not 3–4, days post-hatch in our study), suggesting that there are trade-offs between these behaviors. Our mixed findings regarding effect modification by parental feeding, therefore, might be in part driven by such trade-offs—nests that receive high levels of parent feeding may receive lower levels of brooding compared to nests with low levels of feeding, making it difficult to disentangle what aspects of parental care modify the effects of temperature on nestling growth.

### Limitations and strengths

Some limitations in the sampling design of our study limit our scope of inference and highlight the need for additional studies. First, our sample size may limit our ability to detect weak associations and differences in slopes among strata, which may have led to some of our marginal and non-significant interaction term estimates; further work with larger sample sizes is needed to provide stronger quantitative evidence for these patterns. Second, because our study focused on first broods in a single year, we were unable to investigate whether the patterns we observed are generalizable across broods (first versus later broods) or years. Third, we examined differential effects of temperature across nestling ontogeny by categorizing the nestling period into two time periods, before and after putative development of thermoregulatory independence, based on previous studies of other species of swallows and songbirds [15,53–55]. Further studies are needed to understand how vulnerability to adverse temperatures changes during the gradual development of thermoregulatory independence [55,85]. Fourth, because we were unable to track individual nestlings from hatch, further work is needed to differentiate the effects of age and size in the relationship between relative nestling size and vulnerability to extreme and variable temperatures, such as by manipulating hatching asynchrony [ 94]. Future studies could address many of these limitations by conducting experiments where the variables under consideration (e.g., temperature, brood hierarchy, food delivery) are controlled and manipulated directly by researchers. Finally, it is possible that parental care and/or nestling begging behavior were affected by the collection of morphological and physiological data before our observations, though we attempted to minimize these effects by waiting until birds appeared to resume normal behavior before beginning parental care data collection.

In addition, further research is needed to gain a more complete picture of how variation in temperature and other aspects of weather affect development. We modeled three temperature variables (minimum, maximum, and variability) separately because they capture different aspects of the thermal environment, offering insight on the effects of hot, cold, and variable temperatures on nestlings. Additionally, modeling these variables separately allowed us to avoid issues with collinearity and overfitting our models. However, given that we modeled the three temperature variables separately, we were unable to determine the relative strength of the effects of different temperature variables on nestlings or whether there are cumulative effects of different types of adverse temperature exposures (see discussion in [95,96]). Additionally, various environmental

factors, such as precipitation and wind speed, may have affected the outcome (nestling mass) and relationships we investigated (e.g., see [28,38,97–99]). Although incorporating these additional environmental factors was beyond the scope of our analyses, future studies considering the combined effects of multiple environmental factors would be of great value.

This study also has several strengths. In 113 wild nestling barn swallows, we investigated fine-scale heterogeneity in temperature associations with nestling mass before fledging across ontogeny and social environments, whereas most previous work has focused on characterizing overall patterns in a population or on documenting broader-scale environmental dependence (e.g., across climatic zones). We tracked temperature continuously at the nest-level, rather than at site- or region-level, as many previous studies have done. The fine-scale resolution of our temperature data enabled us to more precisely quantify variation in temperature associations with mass among individuals within the same population, allowing us to investigate the role of developmental timing and early-life social environment in shaping temperature effects on nestlings. Additionally, we characterized feeding levels using detailed measures of parental care behavior collected longitudinally across the nestling period, allowing us to thoroughly characterize average nest-level differences in feeding and assess whether parental feeding can buffer nestlings from the negative impacts of extreme and variable temperatures. We provide a more complete picture of how different aspects of the thermal environment affect nestling growth, while broadening our understanding of how developmental and social factors may confer resilience to or exacerbate the impacts of human-induced environmental change on vulnerable individuals.

## Supporting information

**S1 Fig. Best unbiased linear predictions (BLUPs) for the feeding rate (visits/hour) at each nest across development in wild barn swallows.** A boxplot of feeding BLUPs, measured in visits per hour, is provided for each of two levels of parental care ('low,' 'high') used for stratified analyses in question three.
(PDF)

**S2 Fig. Daily temperatures recorded at wild barn swallow nests in Boulder County, CO.** Daily minimum temperature (A), maximum temperature (B), and temperature variability (interquartile range) (C) in °Celsius recorded by Govee thermometers near each barn swallow nest during the nestling rearing period. Each color corresponds to one of seven breeding sites. Lines connect daily measures for each individual nest.
(PDF)

**S3 Fig. Temperature effects on the mass of 11–13-day-old nestlings at nests receiving low, medium, or high parental feeding.** Predicted relationships of minimum temperature (A), maximum temperature (B), and temperature variability (interquartile range) (C) across the nestling period and nestling mass from linear mixed models, stratified by three levels of parent feeding—low ('Low', n = 35), medium ('Med', n = 36), and high ('High', n = 35). Model predictions are displayed as lines, and raw data are displayed as points. Colors, line types, and shapes correspond to the level of parent feeding. In unstratified analyses (not visualized), there was evidence for an interaction between minimum temperature and parental feeding ($\beta_{med}$ = 1.04 g per 1 SD °C [95% CI: −0.59, 2.53], p = 0.19; $\beta_{high}$ = −1.09 g per 1 SD °C [95% CI: −2.49, 0.28], p = 0.14; F-test for overall effect: p = 0.03). There was no evidence for an interaction between maximum temperature and parental feeding level ($\beta_{med}$ = −0.63 g per 1 SD °C [95% CI: −1.98, 0.66], p = 0.36; $\beta_{high}$ = 0.80 g per 1 SD °C [95% CI: −0.87, 2.48, p = 0.36; F-test for overall effect: p = 0.24) or between temperature variability and parental feeding level ($\beta_{med}$ = −0.80 g per 1 SD °C [95% CI: −2.02, 0.42, p = 0.20; $\beta_{high}$ = 0.84 g per 1 SD °C [95% CI: −0.85, 2.55], p = 0.33; F-test for overall effect: p = 0.15).
(PDF)

**S4 Fig. Photo of a barn swallow perched on a mud cup nest.** This photo, taken during the study, is representative of the features of the nests and nest sites of the birds we monitored.
(PDF)

**S1 Table. Ethogram of barn swallow parental care behaviors.** Each documented behavior has a corresponding location, type (state versus event), description, and developmental context. This ethogram was used for parental care observations. (PDF)

**S2 Table. Background characteristics of wild nestling barn swallows in Boulder County, CO.** Total number of feeding visits, nestling mass, and nestling wing length are separately estimated at three time points (days 3–4, 8–9, and 11–13). Temperature variability is defined as the interquartile range. The table provides the sample size for each variable at the level at which it was collected (n), as well as the mean, standard deviation (SD), minimum, and maximum. (PDF)

**S3 Table. Associations of three temperature variables in early and late development (before or after six days post-hatch) with nestling mass.** Temperature variability is defined as the interquartile range. For each temperature variable, results are provided for unadjusted and adjusted models. Sample size for each developmental stage is provided in the header (n). For each model, the table provides the effect size and 95% confidence interval for temperature effects ($\beta$ (95% CI)), and the corresponding degrees of freedom, t-value, and p-value from a two-tailed t-test using a Satterthwaite degree of freedom estimation. (PDF)

**S4 Table. Associations of three temperature variables with nestling mass, assessed in separate models stratified by relative nestling size at mid development measure (smallest vs. other).** Temperature variability is defined as the interquartile range. For each temperature variable, results are provided for unadjusted and adjusted models. Sample size for each stratum is provided in the header (n). For each model, the table provides the effect size and 95% confidence interval for temperature effects ($\beta$ (95% CI)), and the corresponding degrees of freedom, t-value, and p-value from a two-tailed t-test using a Satterthwaite degree of freedom estimation. (PDF)

**S5 Table. Associations of three temperature variables with nestling mass, assessed in separate models stratified by two levels of parental feeding.** Temperature variability is defined as the interquartile range. For each temperature variable, results are provided for unadjusted and adjusted models. Sample size for each stratum is provided in the header (n). For each model, the table provides the effect size and 95% confidence interval for temperature effects ($\beta$ (95% CI)), and the corresponding degrees of freedom, t-value, and p-value from a two-tailed t-test using a Satterthwaite degree of freedom estimation. (PDF)

**S6 Table. Associations of three temperature variables with nestling mass, assessed in separate models stratified by three levels of parental feeding.** Temperature variability is defined as the interquartile range. For each temperature variable, results are provided for unadjusted and adjusted models. Sample size for each stratum is provided in the header (n). For each model, the table provides the effect size and 95% confidence interval for temperature effects ($\beta$ (95% CI)), and the corresponding degrees of freedom, t-value, and p-value from a two-tailed t-test using a Satterthwaite degree of freedom estimation. (PDF)

## Acknowledgments

We thank Aleea Pardue, Marina Ayala, Heather Kenny-Duddela, Avani Fachon, Grant Gonzalez, and Molly McDermott for assistance and/or advice regarding field work. We thank the private landowners who give us access to their property where barn swallows nest. Finally, we thank members of the Safran, Patricelli, and Karp labs for their constructive feedback on this project.

## Author contributions

**Conceptualization:** Sage A Madden, Rebecca J Safran, Zachary M Laubach.

**Data curation:** Sage A Madden, Zachary M Laubach.

**Formal analysis:** Sage A Madden, Zachary M Laubach.

**Funding acquisition:** Rebecca J Safran, Zachary M Laubach.

**Investigation:** Sage A Madden, Rebecca J Safran, Zachary M Laubach.

**Methodology:** Sage A Madden, Rebecca J Safran, Zachary M Laubach.

**Project administration:** Sage A Madden, Rebecca J Safran, Zachary M Laubach.

**Resources:** Rebecca J Safran, Zachary M Laubach.

**Supervision:** Rebecca J Safran, Gail L Patricelli, Zachary M Laubach.

**Visualization:** Sage A Madden, Zachary M Laubach.

**Writing – original draft:** Sage A Madden, Zachary M Laubach.

**Writing – review & editing:** Sage A Madden, Rebecca J Safran, Gail L Patricelli, Sara R Garcia, Zachary M Laubach.

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
