## [Decision Letter · Decision Letter 0]

19 Nov 2025

Dear Dr. Madden,

We look forward to receiving your revised manuscript.

Kind regards,

Shoko Sugasawa

Academic Editor

PLOS ONE

Journal Requirements:

2. To comply with PLOS One submissions requirements, in your Methods section, please provide additional information regarding the experiments involving animals and ensure you have included details on (1) methods of sacrifice, (2) methods of anesthesia and/or analgesia, and (3) efforts to alleviate suffering.

Reviewers' comments:

Reviewer's Responses to Questions

**Comments to the Author**

1. Is the manuscript technically sound, and do the data support the conclusions?

Reviewer #1: No

Reviewer #2: Yes

2. Has the statistical analysis been performed appropriately and rigorously?

Reviewer #1: No

Reviewer #2: Yes

3. Have the authors made all data underlying the findings in their manuscript fully available?

Reviewer #1: Yes

Reviewer #2: Yes

4. Is the manuscript presented in an intelligible fashion and written in standard English?

Reviewer #1: Yes

Reviewer #2: Yes

Reviewer #1: Overall

The current study focuses on air temperature on nestling body mass in the barn swallow. Authors studied whether the effect of temperature depends on localized early-life developmental constraints imposed by the timing of thermoregulatory development, competition with brood mates, and the amount of parental care, which has received limited attentions. However, the current manuscript suffers at least two issues. The first issue is the lack of explanation. For example, I could not understand why authors compared between the smallest nestlings and others (instead of dividing nestlings into small-sized and large-sized groups). I guess, there must be a good reason to use this classification, but authors should carefully explain the logic. Likewise, I could not understand why authors did not show mean temperature. And I could not understand why authors categorized high/med/low parental care instead of high/low parental care. Authors should also carefully explain correlations between variables. When correlation between variables is high, their relationships with dependent variables might not be independent (and thus they should be analyzed together, rather than separately). Second, authors should analyze interactions between variables. Statistical significance depends on sample size, and thus significant relationship in some groups but not in others does not always mean they differ in slope (e.g., sample size of young nestlings should be larger than that of grown nestlings, sample size of the smallest young should be smaller than that of others, and so on). Thus, to examine whether or not the effects of temperature on body mass depends on some factors, authors need to test the interaction between the grouping factor and temperature on body mass. Or, at the least, authors should explain why they did not test interaction terms. Finally, authors should present their main results in the main text rather than supplements (i.e., all statistical analyses should be shown in the main tables).

Details

L33 the smallest nestling Please see above.

L36 early-life social environment I don’t understand what kinds of “social” environment, you mean here.

L55 key Why? Please explain.

L87 the presence of helpers at the nest What this means? A little bit ambiguous, I think. Please rephrase (and perhaps a little bit in depth explanation is helpful, because authors regard this as a type of “parental” care).

L121 both parents in a social pair provide care Ambiguous expression (what “care” means, here?). Only females brood nestlings, I think.

L126 checked every three to five days Did you accurately estimate hatching asynchrony? I am a little bit skeptical, as five days old nestlings are already big.

L132 temperature near the nest Ambiguous expression.

L151 wing length measures skeletal size Wing length is not “skeletal” size.

L191 low...med...high Please see above. I don’t know why authors used three-level categories. When there are multiple groups, readers should take into account multiple comparison issues (or at least explain whether or not P-value corrections are needed).

L210 likely around six days Ambiguous information.

L226 we balanced our inference What this means? Please explicitly write down.

?238 we z-score Something wrong.

L242 zero variance in some models This is not good reason to exclude site ID from other models.

L269 stratified Please see above. Authors should test interaction terms.

L294 minimum...maximum How about mean temperature? The interpretation of minimum and maximum temperature depends on mean temperature (e.g., imagine max temperature in cold winter days and min temperature in hot summer day: readers do not know the average temperature during the study period in this region).

L305 proportion of time spent brooding This should be high at early nestling stage. I don’t know how authors test the relationship between brooding and feeding (e.g., how did authors control nestling stage using nonparametric methods?). Please explain a little in depth. And please also show sample size in each statistical analysis.

Figs. 3–5 These are analyzed in the same statistics? If not, authors should explain how they statistically assess independent effect of each variable.

L331 did not change substantively What “substantively” means here? Please explicitly explain.

L397 we found some evidence for heterogenous effects What this means? Authors did not use interaction term.

L402 relative nestling size What authors tested is difference between the smallest and other nestlings.

L421 ...and social factors, such as habitat, diet, and nest type Which one is “social”?

L424 more strongly influenced Compared with what?

L431 ontogeny Why authors did not mention about brooding? Brooding should be more frequent in early than in late nestling stages.

L453 limited ability to cool themselves...surface area I think, authors discussion is mainly based on logic (and previous studies) rather than direct observation. Nestlings swallows are brooded by mothers particularly when they are young (and thus low temperature should have particularly large effects when nestlings are young, because parents cannot provide foods while they are brooding). Nestlings open their mouth when they are hot (and thus not only body surface area but relative mouth size would affect thermoregulation). These behaviors should be taken into account when interpreting the main results.

L461 relative size Please see above.

L487 hierarchy But, authors focus solely on the smallest nestlings (i.e., authors did not study size difference among “others”).

L546 unable to track individual nestlings Then, authors cannot discuss about nestling hatching order (e.g., the last hatched nestlings; L474).

L551 Ffith Typo

Fig. 1 I don’t know whether authors could obtain these predictions before having their results (e.g., do two lines cross? If so, where they cross?). Their categorizations (e.g., early/late, smallest/others, low/med/high) needs explanations.

Figs. 3-5 Low-quality figures. Please use high-resolution figures (or vector files such as eps). Please show sample size for each category. Please do not use dot line (because it is often used for non-significant relationships and thus misleading).

Reviewer #2: General

This study investigates the effect of ambient temperature on nestling growth and how its effect depends on the age of the nestlings, brood size, and parental provisioning effort. Including ambient temperature data at each nest in the analysis is an innovative plus. Surprisingly few studies have considered the effect of temperature on different stages of nestling growth combined with parental provisioning effort and this study makes a valuable contribution by doing so.

My only concerns are already enumerated by the authors in their wonderful 'Limitations and strengths' section, so I will mention only that the issue of sample size and the fact that nestlings were handled before rather than after provisioning data were collected could be a potential problem. However, as the authors clearly identify these limitations, it will make it easy for readers to take them into account.

Specific

As PLOS One does not copy-edit, I have listed the typographical/grammatical errors that I noticed.

line 43: hyphenate 'At risk'

line 91: Add scientific name after barn swallow

line 120: Add 'external' before morphological

line 124: hyphenate 'first brood'

line 145: clarify if two blood samples were collected at each of the two sample times or if one sample was collected at each of

the two times

line 513: separate 'maybe' into two words

line 551: replace 'Ffith' with Fifth.

lines 552-555: Add statement that handling and drawing a blood sample may also have affected nestling begging behavior

References: format needs to be standardized

.

Reviewer #1: No

Reviewer #2: No

---

## [Author Response · Author response to Decision Letter 1]

31 Dec 2025

Point-by-point response to the editor’s and reviewers’ comments (also in attached "Response to Reviewers" document)

Editor

Comment. Thank you for submitting your manuscript to PLOS ONE. After careful consideration, we feel that it has merit but does not fully meet PLOS ONE’s publication criteria as it currently stands. Therefore, we invite you to submit a revised version of the manuscript that addresses the points raised during the review process.

While both reviewers agree that this study addresses an important knowledge gap, reviewer 1 requested more details on methodology. In particular, please: provide the rationale for comparison between the smallest nestlings and others (instead of dividing nestlings into small-sized and large-sized groups); and describe correlations and interactions between variables if any. Additionally, it would be great if you could address their query on minor technical points.

Response. Thank you for your comments. Below and in the revised text, we address these suggestions.

Reviewer #1

Comment. Overall: The current study focuses on air temperature on nestling body mass in the barn swallow. Authors studied whether the effect of temperature depends on localized early-life developmental constraints imposed by the timing of thermoregulatory development, competition with brood mates, and the amount of parental care, which has received limited attentions. However, the current manuscript suffers at least two issues. The first issue is the lack of explanation. For example, I could not understand why authors compared between the smallest nestlings and others (instead of dividing nestlings into small-sized and large-sized groups). I guess, there must be a good reason to use this classification, but authors should carefully explain the logic. Likewise, I could not understand why authors did not show mean temperature. And I could not understand why authors categorized high/med/low parental care instead of high/low parental care. Authors should also carefully explain correlations between variables. When correlation between variables is high, their relationships with dependent variables might not be independent (and thus they should be analyzed together, rather than separately). Second, authors should analyze interactions between variables. Statistical significance depends on sample size, and thus significant relationship in some groups but not in others does not always mean they differ in slope (e.g., sample size of young nestlings should be larger than that of grown nestlings, sample size of the smallest young should be smaller than that of others, and so on). Thus, to examine whether or not the effects of temperature on body mass depends on some factors, authors need to test the interaction between the grouping factor and temperature on body mass. Or, at the least, authors should explain why they did not test interaction terms. Finally, authors should present their main results in the main text rather than supplements (i.e., all statistical analyses should be shown in the main tables).

Response. Thank you for your thorough review and comments indicating where additional information or clarification is needed. We have responded to each of the issues you’ve raised below.

Issue 1. The first issue is the lack of explanation. For example, I could not understand why authors compared between the smallest nestlings and others (instead of dividing nestlings into small-sized and large-sized groups). I guess, there must be a good reason to use this classification, but authors should carefully explain the logic.

Response. We now include an explanation of why we chose to compare the smallest nestling to other nestlings (see below).

Pg 8, Lines 152-156: We choose to compare the smallest nestling to all other nestlings because, in North American barn swallows, incubation generally begins when the penultimate egg is laid [32]; As a result, it is common for one nestling to hatch ~one day later than its brood mates, and we expected the largest size disparity to be between this last hatched nestling and all others [33].

Issue 2. Likewise, I could not understand why authors did not show mean temperature.

Response. Thank you for bringing this to our attention. We now include the mean temperature during the nestling period in the “Background characteristics” section and S2 Table.

Pg 15, Lines 317-320: For the near-nest temperature measurements averaged across the nestling period, the mean ± SD was 23.99 ± 1.32 degrees C for mean temperature, 13.10 ± 1.80 degrees C for minimum temperature, 36.90 ± 2.96 degrees C for maximum temperature, and 9.26 ± 2.43 degrees C for temperature variability.

Issue 3. And I could not understand why authors categorized high/med/low parental care instead of high/low parental care.

Response. We now present the results of analyses where parental care is stratified into two levels (high / low). We additionally include the three-level analysis (high / med / low) as a sensitivity analysis to assess the robustness of our results to differences in the selected number of strata for parental feeding. The methods, results, and discussion have been updated to reflect the changes in analyses.

Issue 4. Authors should also carefully explain correlations between variables. When correlation between variables is high, their relationships with dependent variables might not be independent (and thus they should be analyzed together, rather than separately).

Response. We assessed the effects of minimum, maximum and variability of temperature separately based on our a priori hypothesis that these conditions would have separate effects on nestling growth. While we expect a correlation between minimum and maximum temperature, our question was not to assess the effect of each temperature extreme while holding constant the other temperature values. This approach also allows us to avoid issues with collinearity and overfitting our model. We have clarified the rationale for and limitations of modeling the three temperature variables separately in the “Limitations and Strengths” section (see below)

Pg 29-30, Lines 633-640: Second, we modeled three temperature variables (minimum, maximum, and variability) separately because they capture different aspects of the thermal environment, offering insight on the effects hot, cold, and variable temperatures on nestlings. Additionally, modeling these variables separately allowed us to avoid issues with collinearity and overfitting our model. However, given that we modeled the three temperature variables separately, we were unable to determine the relative strength of the effects of different temperature variables on nestlings or whether there are cumulative effects of different types of adverse temperature exposures (see discussion in [90,91]).

We chose to run separate models for our three questions (focused on developmental stage, relative nestling size, and parental feeding level) because we were interested in the overall effect of each of these variables, rather than the effect of each variable after adjusting for the other variables (which is what a model of all three variables together would tell us). Additionally, based on our study design, we did not expect correlations among these three variables (e.g., parental feeding is measured at the nest level, and we would not expect any correlation of parental feeding with relative nestling size because, excepting occasional mortality, all nests contain “min” and “other” nestlings).

Issue 5. Second, authors should analyze interactions between variables. Statistical significance depends on sample size, and thus significant relationship in some groups but not in others does not always mean they differ in slope (e.g., sample size of young nestlings should be larger than that of grown nestlings, sample size of the smallest young should be smaller than that of others, and so on). Thus, to examine whether or not the effects of temperature on body mass depends on some factors, authors need to test the interaction between the grouping factor and temperature on body mass. Or, at the least, authors should explain why they did not test interaction terms.

Response. For questions concerning effect modification, we now include models with an interaction term between temperature and the effect modifier, in addition to stratified analyses (see updated “Methods” text below).

Pg 13-14, Lines 282-292: To determine whether effect modification was present, we ran analyses stratified by multiple levels of relative nestling size or parental feeding. The results from these stratified models provide insight into the magnitude and directionality of the effect of temperature on nestling mass for different relative nestling sizes or levels of parental feeding. In addition, for each question, we ran a model with unstratified data including an interaction term between temperature and the effect modifier (relative nestling size or parental feeding). The estimate, confidence interval, and p-value for the interaction term were used to inform interpretation of stratified results. We present stratified results even in the absence of significant interaction terms, given that interaction terms can be insignificant when effect modification is present due factors such as low power, the scale of measurement selected, and presence of non-linear relationships [50–53].

Issue 6. Finally, authors should present their main results in the main text rather than supplements (i.e., all statistical analyses should be shown in the main tables).

Response. The main results are reported in-text and represented in the figures, making the model result tables redundant. We are concerned that including the model result tables in the main text may create confusion, as readers may wonder why we present the same results several times. For these reasons, we prefer to keep the main tables in Supplemental Material. However, if the Reviewer and Editor disagree, we can add the model results tables to the main text.

Comment. L33 the smallest nestling Please see above.

Response. Please see our response to the overall comment above (under “Issue 1”).

Comment. L36 early-life social environment I don’t understand what kinds of “social” environment, you mean here.

Response. We clarified the wording of this sentence.

Pg 2, Lines 34-37: These findings indicate the existence of fine-scale heterogeneity in which the effects of temperature on nestling development are sensitive to metabolic constraints and early-life social environment (e.g., size relative to siblings), providing insight into the factors that may ameliorate or exacerbate climate impacts on individual birds.

Comment. L55 key Why? Please explain.

Response. We clarified the wording of this sentence.

Pg 3, Lines 54-56: This knowledge would advance our understanding of heterogeneity in susceptibility of nestling birds to climate impacts and the factors that may ameliorate or exacerbate these impacts.

Comment. L87 the presence of helpers at the nest What this means? A little bit ambiguous, I think. Please rephrase (and perhaps a little bit in depth explanation is helpful, because authors regard this as a type of “parental” care).

Response. We clarified the wording of this sentence.

Pg 5, Lines 85-87: Similarly, studies of cooperatively breeding bird species suggest that the presence of helpers—additional, non-parent individuals that care for offspring (an example of alloparental care)—may mitigate negative impacts of weather on nestlings [18] (but see [29]).

Comment. L121 both parents in a social pair provide care Ambiguous expression (what “care” means, here?). Only females brood nestlings, I think.

Response. We clarified the wording of this sentence.

Pg 6, Lines 121-122: In this socially monogamous species, both parents in a social pair provision food until nestlings fledge (>18 days post hatch) [32].

Comment. L126 checked every three to five days Did you accurately estimate hatching asynchrony? I am a little bit skeptical, as five days old nestlings are already big.

Response. We believe our measure of hatch asynchrony to be accurate for several reasons. First, once nests were close to their estimated hatch date, we checked them every two days—we added text clarifying our nest monitoring methods (see below). Second, aging of nestlings was based on a previous study (cited in-text, see below) which provided detailed descriptions of developmental characteristics present in nestling barn swallows at each age. Taken together, our monitoring efforts and reliable estimation of newly hatched nestlings’ age allowed accurate assessment of hatch asynchrony.

Pg 6-7, Lines 126-131: Nests were checked every three to five days to track nestling phenology and fate, and when nests were close to their estimated hatch date (based on clutch initiation date), they were checked every two days. During the first check after nestlings hatched, we estimated their ages (hatch day = day zero) based on feather emergence and other reliable developmental characteristics, such as wetness (on hatch day) and ability to raise their head [35].

Comment. L132 temperature near the nest Ambiguous expression.

Response. We clarified the wording of this sentence.

Pg 7, Lines 133-134: We monitored temperature using loggers placed 10-28 cm from the nest from hatch through last nestling measures (~ day 12).

Comment. L151 wing length measures skeletal size Wing length is not “skeletal” size.

Response. We removed this phrase.

Pg 8, Lines 156-157: Our classification was based on right wing length, rather than mass, because wing length may not be as strongly influenced by short-term fluctuations as mass [37–39].

Comment. L191 low...med...high Please see above. I don’t know why authors used three-level categories. When there are multiple groups, readers should take into account multiple comparison issues (or at least explain whether or not P-value corrections are needed).

Response. We now include analysis of a two-level parental care variable. Please see our response to the overall comment above (under “Issue 3”).

We did not correct for multiple comparisons to account for multiple groups because the goal of this analysis was to assess consistency, or heterogeneity, of associations between temperature and growth among subgroups given the expectation that these effects have similar biological underpinnings. Accordingly, such correction would be overly stringent and has potential to inflate type 2 error.

Comment. L210 likely around six days Ambiguous information.

Response. We clarified the wording of this sentence

Pg 10, Lines 210-214: We chose these time periods based on previous research in swallows and other songbirds [14,44–46], which found that nestling swallows begin to develop the ability to thermoregulate independently around four to five days post-hatch, and the age of the effective homeothermy (defined here as ability to maintain relatively constant body temperature under natural conditions—in the nest with brood mates) is approximately six days of age.

Comment. L226 we balanced our inference What this means? Please explicitly write down.

Response. We removed ambiguous wording and explicitly defined how we report and interpret results.

Pg 12, Lines 239-244: We focused our reporting and interpretation of results on estimates of effect sizes and 95% confidence intervals, providing information on the strength and uncertainty of our effect estimates [47], as well as p-values interpreted using the language of evidence [48]. P-values less than or equal to 0.05 were interpreted as providing moderate to strong evidence for an effect, p-values between 0.05 and 0.10 as providing marginal evidence, and p-values greater than 0.10 as providing little to no evidence [48].

Comment. L238 we z-score Something wrong.

Response. We clarified the wording of this sentence.

Pg 12, Lines 249-251: For all models, we z-score standardized (subtracted the mean and divided by the standard deviation) numeric explanatory variables and covariates to aid in model fit and comparison of effect sizes.

Comment. L242 zero variance in some models This is not g

---

## [Decision Letter · Decision Letter 1]

25 Jan 2026

Dear Dr. Madden,

Thank you for submitting your manuscript to PLOS ONE. After careful consideration, we feel that it has merit but does not fully meet PLOS ONE’s publication criteria as it currently stands. Therefore, we invite you to submit a revised version of the manuscript that addresses the points raised during the review process. See comments at the end of the message for specifics.

We look forward to receiving your revised manuscript.

Kind regards,

Christopher A. Lepczyk

Academic Editor

PLOS One

Journal Requirements:

Additional Editor Comments:

Overall, this ms is an interesting evaluation of the relationship between food provisioning and temperature on the growth of altricial birds. Having taken over this ms from the previous editor, I have read through the previous comments and responses. Both reviewers found merit in the work, with Reviewer 1 still having a number of comments to consider and reviewer 2 being satisfied. Most of Reviewer 1’s comments are straightforward to address and will improve flow of the ms. The issue of p-values is one that many scientists are on different views about and thus while you provided a citation supporting your approach, it was clear the reviewer does not agree with that view. In any case, as long as you have complete p-values shown, then I concur with your approach (though I would note that if you can provide a measure of fit or explanatory power that is helpful). Finally, I have a number of comments related to altricial bird growth and the environment that I listed below and will help to clarify your work. I look forward to seeing a revised ms.

L46. I would suggest describing species not strictly as passerine songbirds, but as altricial species. The topic you are interested in less specific taxonomic distinction and more on the developmental continuum.

L50. How is temperature affecting food delivery? I would suggest separating out these ideas and explaining more. Precipitation or marked decrease in temperature are often what drives changes in food delivery.

L57. In altricial birds.

L58. I would suggest adding in approximate days during which you consider young for a small passerine. For most altricial species this true early period (and for your species of study) is hatch to several days. But help the reader understand more about how you define time during the in nest growth period.

L61. One item I’m a bit unclear here is are you meaning temperature increase or decrease or both? The reason that this is important as it often varies over the breeding season with early spring having bouts of cold snaps and later into summer having potential heat wave. Also, precipitation is often the other variable driver here of food provisioning rates and at least should be mentioned.

L63. Same comment as above. Give some approximation of percent of growth completed or number of days to reach this benchmark.

L69. Some, but most nestlings are born relatively close to together and typically have one that does not. How far apart in time does asynchrony matter here?

L81. Is it parental care or is it really just food provisioning or one of the parents sitting on the nest?

L93. Figure acknowledgement should be moved to your hypotheses as it isn’t showing the questions you are testing, but the hypotheses.

L100. I would suggest that you move the predictions out of the figure and that they follow on their respective hypothesis. Also, for each of the predictions I would expect to see citations or deductive reasoning as to why you expect to see these relationships.

L103. I would suggest just one figure that has the multiple panels be in main body of ms. No need for this sentence explaining that information.

L107. Cut this sentence from the figure legend.

L114. One limitation of this work in relation to climate change is that you are studying only a single year. I would suggest that the key part of your work is really on intra-annual variability as that was all you are really going to be able to do.

L117. Please give the mean and SD of clutch size.

L119. Since you have two negative statements here I would suggest using the words ‘neither’ and ‘nor.’

L112. Give mean and SD for fledging in this species, ideally in your study location.

L125. How was this timing of visitation decided? Five days is quite long for altricial nestling monitoring in relation to what you are studying.

L133. Outside of the nest?

L137. Cut this explanation sentence.

L146. How were able to differentiate individuals before this point in time and isn’t that an important part of the study?

L147. How was blood collected and how much?

L149. There are methods to tag/mark individuals younger than this age. Unclear why that wasn’t done.

L159. Just curious why a lab experiment wasn’t attempted to actually control a lot of these pieces of information? The inexact aging here and lack of telling individuals apart is a big limitation.

L165. Is there precedent for monitoring after you’ve done the measurements rather than beforehand? I would have thought that beforehand would be better.

L177. Given this situation why not just always use cameras?

L183. Need to indicate you are using r.

L200. Ok, but already one struggle I’m having is how much of the data you are collecting is simply being put into categorical terms and not continuous. Growth and regulation is fundamentally a continuous process and much is being missed by a categorical approach.

L215. Did you measure days of inclement weather over the course of the study? Precipitation directly affects many aerial insectivores and thus feeding rate in your study and hence growth could easily be affected by it as much or more so than temperature.

L241. Did you also calculate any measure of fit? Helpful to readers to see not just p-values and CI’s.

L264, L273, L298, L419, L570, L586. I would suggest changing the wording here as sensitivity analysis is a formal comparison in simulation modeling of parameter variation. You didn’t quite do that. Change wherever this issue occurs.

L297, L331-3. Avoid stating a figure for reader to look up and instead explain what you mean and have the figure notation in parentheses.

L318. Use degree symbol throughout.

L320. Can cut as this is restating what you did. I would suggest cleaning up the Results section further as noted by the reviewer. Do not need to describe what you did or how you did it, just main findings and reporting of stats.

L326. I would suggest that you end the stats section in the Methods by saying something like ‘All results are presented as means +/- SD, unless otherwise stated.’ This statement will cut some wording here and make it clear throughout the Results what is being reported.

L354, L440. Move this sentence up to previous paragraph and add it in. Paragraphs should be three sentences or more in length.

L443. Cut this first paragraph and replace it with whether or not you found support for each hypothesis. Do not need to restate the reason for the research or sample size, etc.

L483. And precipitation.

L486, L503, L522. Move up to beginning of Discussion.

L505, L513, L556. Need an object after the word ‘this’ so there is logical connection to previous sentence. Double check ms throughout for issue.

L561. Reviewer 1 suggested a shorter Discussion and I concur. This section is one where you could cut down the length and just give succinct reasons, rather than some of the back and forth.

L594. I would recommend that some of these studies be done in the lab where most of the factors can be controlled. It is possible to raise nestlings in the lab and vary temp, humidity, food delivery, etc. and monitor individuals.

L631. For a field based growth study you have a pretty solid sample size. Yes, more is better, but I think you want to consider other reasons. Timing in the year when the work was done, effects of precipitation, if relationships vary year to year, etc. I think you are spending a lot of time on issues that are important, but could be mitigated by a more elegant lab based approach (or even better a lab-field combined study).

L671. Can cut this last section, no need to just restate findings. If there are implications of your work, that would be a different way to end the work.

Reviewers' comments:

Reviewer's Responses to Questions

**Comments to the Author**

Reviewer #1: (No Response)

Reviewer #2: All comments have been addressed

2. Is the manuscript technically sound, and do the data support the conclusions?

Reviewer #1: Partly

Reviewer #2: (No Response)

3. Has the statistical analysis been performed appropriately and rigorously?

Reviewer #1: No

Reviewer #2: (No Response)

4. Have the authors made all data underlying the findings in their manuscript fully available?

Reviewer #1: Yes

Reviewer #2: (No Response)

5. Is the manuscript presented in an intelligible fashion and written in standard English?

Reviewer #1: Yes

Reviewer #2: (No Response)

Reviewer #1: Overall

Thank you very much for your revision. The revised manuscript is better than the initial manuscript. However, I have some comments on the revised manuscript. First of all, authors should not regard non-significant results as statistical “support.” Only when authors had significant results (i.e., P < 0.05), authors can obtain statistical support. All descriptions using marginal (i.e., 0.05 < P < 0.10) as a support for the hypothesis should be deleted (or at the least, should not regarded as a support for the hypothesis). Authors should be objective throughout the text. Authors should also carefully write down their statistics (e.g., test statistics of interaction terms). Second, Authors’ discussion is too long. Rather than explaining each result one by one, authors should discuss overall patterns (based on objective statistical results). Please see below for each comment.

L21 avoid exposure Unclear (=ambiguous) expression. Please rephrase.

L26 mass Body mass would be better, I guess.

L29 higher minimum temperature were associated with lower nestling mass Really? I think, “associated with higher nestling mass” would be a correct expression here. Please check.

L31 marginal Please see above.

L46 development of adult phenotype Authors should explain why this is important (reproduction and later survivorship, right?).

L58-L63 These expressions look redundant (i.e., authors used similar expressions multiple times). Please revise.

L65 most studies... as a whole, leaving unanswered...before versus after Unclear expression, I guess. Authors can change expression so that readers can easily contrast previous studies and the current study (i.e., To study the importance of developing thermoregulation, we don’t need to compare before vs after, e.g., including days after hatching as an independent variate, and thus the current expression would be misleading).

L83 burying beetles This is an insect species, and thus authors should carefully and briefly explain why this example can be used for understanding birds.

L87 weather Do this “weather” means temperature? If so, please write it down. If not (e.g., rain), I am not sure why authors cite this article. Please explain.

L101 advantageous social environments (e.g., larger size Authors focus on “the smallest” (L96) and thus authors should focus on the smallest (rather than “others”) throughout the text.

L115 Boulder Please provide latitude, longitude, altitude.

L138 and others ~six days Please use English expression rather than “~”.

L154 ~one day Please see above.

L167 Observation Please provide weather information. For example, authors did observation at rainy or windy day? Likewise, throughout the text, temperature and weather (e.g., rainy, windy) should be separated.

L180 generalized linear mixed effects model Please explain detailed information of the statistics (e.g., statistical program used). When using negative binomial distribution, authors should explain how they control for overdispersion (which is automatically corrected in some programs but not in others).

L180 total observation time Please provide mean, min, and max observation time.

L185 number of days since the first nestling hatched Did authors assume that provisioning rate increase linearly with days? This might not be the case (please see L328). Please explain.

L216 over the entire nestling period How this information is used? Please explain (because I assume that authors focused on temperature information before and after key date).

L242 equal to 0.05 P = 0.05 is marginal (i.e., P < 0.05 is significant).

L271 mass on days 11-13 was the outcome Then, the effect of temperature on young nestlings would be diluted, which might affect the results.

L290 insignificant -> nonsignificant

L318 degrees C Do these values include night-time temperature? Please explain (because female swallows sit on the nests without feeding during night, night-time and day-time temperature should be distinguished). Also, please use °C throughout the text (as in L336).

L330 Spearman rank correlation This is wrong (as authors told that this analysis is conducted throughout the nestling period in their reply to my previous comment). Imagine that parents incubate more and needs less foods during early nestling period than during late nestling period. Then, without tradeoff between the two parental care behaviors (i.e., brooding and provisioning), there can be a negative relationship between them (because of nestling period as a confounding factor). Authors should control the confounding factor when focusing on the relationship between brooding and provisioning.

L331 p = Why authors used lowercase p here (but used uppercase P in others)? Please revise.

L336 β = ... [95% CI: Please provide SE and test statistics (t) throughout the results section. I am also not sure why authors used bootstrap to estimate 95%CI. Because authors used t-values for estimating P-values, authors had SE values as well and can estimate 95% CI based on them.

L344 Please explain why authors could not test interactions here (so that readers can easily understand the reason).

L355 substantially Please rephrase “substantially” with objective statistical terminology. For example, significant and non-significant relationship remains unchanged? What is “substantial” differs between researchers (and thus can be somewhat subjective) and thus authors should avoid using these kinds of words. I also did not understand what “precision” here means.

L364 and others interaction... β = Authors should test the interaction term (e.g., by comparing likelihoods of models with and without interaction terms), rather than testing the difference in slope in the model with interaction term.

L365 marginal support Please see my comment above.

L373 β = 0.37 Why authors did not use unit here? Please describe results in the same manner throughout the text.

L373, 374, 395 and others was (...) Please revise. Please do not use parenthesis like that.

L385 only “Smallest” or “Other” nestling points are present for some nests. Then, authors should exclude these nests, as they experience changing social environment (i.e., not comparable with other nests in which social environments were unchanged).

L428 Please provide statistics for interaction terms here.

L433 were associated with ... P = 0.07 P = 0.07 is not significant.

L441 substantially Please see above.

L442 Discussion Too long.

L456 was inconclusive, but... may be suggestive of This expression looks somewhat subjective (because authors stress positive results alone). Authors should also explain multiple testing. It is not surprising that some of many tests had “significant” results by chance.

L465-473 This part looks redundant. Please revise.

L478 weather I guess, authors used weather as a synonym to temperature, but they are not identical. Authors should distinguish weather (e.g., sunny, cloudy, and so on) and temperature, even if they are correlated.

L508 Early in development...Late in development This contrast looks strange because authors found similar pattern regardless of developmental period (L503). Authors should explicitly explain why they found similar pattern regardless of the contrasting thermoregulatory issues between early and late developmental periods. It should also be noted that each nest might experience similar temperature condition between early and late nestling periods (and thus the observed pattern would be cumulative effects rather than independent effects of temperature in the each nestling period).

L522 marginal Please see above.

L531-L541 This paragraph resembles with L465-473. Please do not explain each result one by one. Authors should explain overall pattern in the discussion.

L557 relatively small Please explain why authors think that the difference between 26.8 and 32.4 was small (for me, >15% difference is large).

L576 p-values were above the cut-off for marginal evidence Please note that cut-off point is P = 0.05.

L587 may indicate effect modification, even in the absence of a significant interaction term This is not good. Authors should objectively judge their results based on statistics and clear logic (please do not subjectively pick up your favorite results while ignoring others).

L594 non-linear relationships Then, instead of using categorization, authors can directly use estimate of provisioning rate across nestling period.

L609 negatively corelated, suggesting that there are trade-offs Please see above.

L611 nests that receive high levels of parent feeding may receive lower levels of brooding This is not shown (statistics across developmental period does not clarify patterns across nests in the same developmental period).

L615-L628 This paragraph is hard to understand. Please rewrite.

L648 degrees of hatch asynchrony might be related to nest temperature during incubation This is beyond the scope of the current study. Also, authors should use “hatching asynchrony” instead of “hatch asynchrony.”

L673 differed among three temperature variables This is not surprising.

Please use vector files.

Fig. 2 I like illustrations in this figure, but illustration of fledging swallows should be revised (because fledglings do not have long tails). It would be better to add the exact days when they fledge.

Fig. 3-5 Please do not use broken lines.

Supplementary Tables I am not sure why authors would like to put these tables in supplementary materials. Although authors respond to my previous comment that tables are redundant to in-text statistics, but then, authors should use tables rather than in-text statistics. Readers can easily access to statistics to tables (rather than in-text statistics). And authors should use “standard” table styles used by other articles.

Statistics Authors did not report correlation between variables. This is not good. For example, how temperature variables in the early nestling period are correlated with those in the late nestling period? If they are tightly correlated (e.g., nests in variable temperature environment in the early nestling period experience variable temperature in the late nestling period, too), it is not surprising that similar pattern can be found in early and late nestling periods. Of course, authors know the pattern, and thus they should report correlations among variables (and explain why authors did not statistically control for correlations among variables). I am also not sure why authors did not include these variables in the same model.

Reviewer #2: (No Response)

.

Reviewer #1: No

Reviewer #2: No

---

## [Author Response · Author response to Decision Letter 2]

6 Mar 2026

Point-by-point response to the editor’s and reviewers’ comments

Editor

Comment. Overall, this ms is an interesting evaluation of the relationship between food provisioning and temperature on the growth of altricial birds. Having taken over this ms from the previous editor, I have read through the previous comments and responses. Both reviewers found merit in the work, with Reviewer 1 still having a number of comments to consider and reviewer 2 being satisfied. Most of Reviewer 1’s comments are straightforward to address and will improve flow of the ms. The issue of p-values is one that many scientists are on different views about and thus while you provided a citation supporting your approach, it was clear the reviewer does not agree with that view. In any case, as long as you have complete p-values shown, then I concur with your approach (though I would note that if you can provide a measure of fit or explanatory power that is helpful). Finally, I have a number of comments related to altricial bird growth and the environment that I listed below and will help to clarify your work. I look forward to seeing a revised ms.

Response. Thank you for your detailed and helpful comments on altricial bird growth and the environment. Additionally, thanks for your acknowledgement of the many different views surrounding p-values. We respond to each of your line-by-line comments below.

When working on revisions, it came to our attention that there was a mismatch in our statistical approach with that of a previous publication with the same set of authors. Specifically, the mismatch was in the unstratified model looking at the interactive effects of parental feeding BLUPs and temperature on nestling mass. In this MS, we had originally modeled the interaction using a continuous variable for parental feeding BLUPs. However, in the previous publication considering effect modification of the relationship of growth and physiological stress response by parental care (Laubach et al. 2025 American Naturalist), we instead modeled the interaction using a categorical variable for parental feeding level, where feeding BLUPs were divided into ‘high’ and ‘low’ categories. This categorical interaction matches more closely with the stratified analyses in both manuscripts, where we ran a separate model for each of the two parental feeding levels (high and low). As such, in this MS, we now model the interaction using a two-level categorical variable for parental feeding level. This choice ensures consistency with similar analyses in a previous publication and a better match of our unstratified and stratified analyses. Additionally, this approach is more consistent with our analysis for question two, in which we looked at the interactive effects of a categorical variable for relative nestling size and temperature on nestling mass.

Comment. L46. I would suggest describing species not strictly as passerine songbirds, but as altricial species. The topic you are interested in less specific taxonomic distinction and more on the developmental continuum.

Response. Done.

Comment. L50. How is temperature affecting food delivery? I would suggest separating out these ideas and explaining more. Precipitation or marked decrease in temperature are often what drives changes in food delivery.

Response. We now explain these ideas further in the Introduction paragraph focused on parental care.

Pg 5, Lines 97-104: Finally, adverse weather conditions may lead to reduced food provisioning. For instance, provisioning rates may decrease in hot or cold temperatures due to reduced availability of insect prey [11,32] and parents diverting more time and energy to their own thermoregulation [33,34]. Similarly, food availability and provisioning rates often decrease during periods of precipitation or high wind, especially for aerial insectivores [29,35–38]. Therefore, it is plausible that parental feeding influences nestling vulnerability to adverse conditions, including extreme temperatures, in altricial birds.

Additionally, we have added a discussion of the potential impacts of precipitation and other environmental conditions on the variables and relationships under study to the “Limitations and Strength” section.

Pg 28, Lines 600-602: Second, because our study focused on first broods in a single year, we were unable to investigate whether the patterns we observed are generalizable across broods (first versus later broods) or years.

Pg 29, Lines 625-629: Additionally, various environmental factors, such as precipitation and wind speed, may have affected the outcome (nestling mass) and relationships we investigated (e.g., see [28,38,97–99]). Although incorporating these additional environmental factors was beyond the scope of our analyses, future studies considering the combined effects of multiple environmental factors would be of great value.

In our study, we were interested in the extent to which varying levels of parental feeding summarized across development, differentially affected nestling vulnerability to extreme temperatures, rather than the effects of temperature on food provisioning itself. As such, we accounted for the potential effects of temperature on food delivery when creating the index of overall parental feeding levels and treated this variable as an effect modifier in our analyses.

Pg 10-11, Lines 211-226: We created an index of the level of parental feeding by summarizing feeding rates from both parents across all stages of nestling development. Specifically, we used a generalized linear mixed effects model, in which total feeding count by both parents measured at each developmental stage was the outcome and nest ID was a random intercept. To fit models, we used the ‘lme4’ package, version 1.1.35.5 [49] in R version 4.4.1 [50]. We compared models with several error distributions and link functions using diagnostic plots and tests from the package ‘DHARMa,’ version 0.4.6 [51]. We selected a negative binomial distribution and log link for our final model because this appropriately handled overdispersion in the count data. The model included an offset for total observation time and several covariates that could influence parental feeding behaviors: number of days since the first nestling hatched, number of nestlings in the nest, the median temperature near the nest during the observation period, and the duration of time between removing nestlings from the nest for morphometric and physiological measures and the start of the observation. From this model, we extracted the best linear unbiased predictors (BLUPs) (following [43]). Because parental feeding is an effect modifier in our models, we created a two-level categorical variable for stratified analyses by classifying BLUPs as ‘low’ (lowest half) or ‘high’ (highest half) parental feeding (S1 Fig).

Comment. L57. In altricial birds.

Response. Done.

Comment. L58. I would suggest adding in approximate days during which you consider young for a small passerine. For most altricial species this true early period (and for your species of study) is hatch to several days. But help the reader understand more about how you define time during the in nest growth period.

Response. We have removed the word “young” and instead focus on the timing of thermoregulatory development during the nestling period, as well as the potential effects of temperature on nestlings before or after endothermy develops.

Pg 3-4, Lines 60-72: The effects of temperature on altricial nestling development may differ depending on the timing of exposure during development [11–14]. During the first few days after hatch, nestlings are mostly featherless and lack the ability to thermoregulate independently [8,15]. Consequently, exposure to hot or cold ambient temperatures may drive body temperatures outside of the narrow range optimal for growth [5,6,11,13]. Endothermy typically develops over several days in the middle of the nestling period [6,15], with timing influenced by growth rate of the species, brood size, and a variety of other factors [5,15,16]. After endothermy develops, nestlings exposed to hot or cold temperatures may expend more energy on thermoregulation and/or enter a state of hyperthermia (in the case of extreme heat), leading to increased stress and decreased growth [6,17,18]. Because the thermoregulatory abilities of nestlings change over the course of the nestling period, we might expect the effects of temperature on nestling growth to vary across ontogeny. This possibility warrants further investigation, as most previous studies have focused on the nestling period as a whole (but see [11–14]).

Comment. L61. One item I’m a bit unclear here is are you meaning temperature increase or decrease or both? The reason that this is important as it often varies over the breeding season with early spring having bouts of cold snaps and later into summer having potential heat wave. Also, precipitation is often the other variable driver here of food provisioning rates and at least should be mentioned.

Response. We now specify that we are interested in the effects of exposure to both cold and hot temperatures.

Pg 3-4, Lines 60-69: The effects of temperature on altricial nestling development may differ depending on the timing of exposure during development [11–14]. During the first few days after hatch, nestlings are mostly featherless and lack the ability to thermoregulate independently [8,15]. Consequently, exposure to hot or cold ambient temperatures may drive body temperatures outside of the narrow range optimal for growth [5,6,11,13]. Endothermy typically develops over several days in the middle of the nestling period [6,15], with timing influenced by growth rate of the species, brood size, and a variety of other factors [5,15,16]. After endothermy develops, nestlings exposed to hot or cold temperatures may expend more energy on thermoregulation and/or enter a state of hyperthermia (in the case of extreme heat), leading to increased stress and decreased growth [6,17,18].

In addition, we provide information on potential effects of precipitation on food provisioning in the paragraph focused on parental care—please see our response to the comment on L50.

Comment. L63. Same comment as above. Give some approximation of percent of growth completed or number of days to reach this benchmark.

Response. Our understanding is that the timing of development of endothermy is variable within and across species. We now explain this with more nuance and provide a rough approximation of when during the nestling period endothermy usually develops. Please see our response to the comment on L58.

Comment. L69. Some, but most nestlings are born relatively close to together and typically have one that does not. How far apart in time does asynchrony matter here?

Response. Thank you for asking this question. We have revised this section to focus on hatching asynchrony where some nestlings hatch a day or two later than their brood mates, as this is a common pattern in altricial birds and was the focus of our study.

Pg 4, Lines 74-78: [19,20]. Altricial nestlings often hatch asynchronously, with some nestlings hatching a day or two later than their brood mates, as a result of incubation beginning before all eggs are laid [21]. Hatching asynchrony typically results in size asymmetry, with younger nestlings remaining smaller than their brood mates across development [21].

Comment. L81. Is it parental care or is it really just food provisioning or one of the parents sitting on the nest?

Response. We now include food provisioning as an example of parental care. Our study focuses on food provisioning which provides a direct resource benefit to nestlings. We do not focus on brooding or other parental care behaviors in this study, expect where they aid our inferences about feeding behavior.

Pg 5, Lines 87-92: Parental care, including food provisioning, is another aspect of the social environment that may shape nestling responses to adverse conditions [19,20,30]. Nestlings are entirely dependent on food provisioned by parents to provide energy for growth and thermoregulation. As such, nestlings that receive more parental care may be less susceptible to the negative effects of temperature and other adverse conditions because they have more energetic reserves for thermoregulation and growth.

Comment. L93. Figure acknowledgement should be moved to your hypotheses as it isn’t showing the questions you are testing, but the hypotheses.

Response. Based on another comment, we now provide the predictions in the main text and acknowledge the figure at that time.

Comment. L100. I would suggest that you move the predictions out of the figure and that they follow on their respective hypothesis. Also, for each of the predictions I would expect to see citations or deductive reasoning as to why you expect to see these relationships.

Response. We now provide the predictions in-text, following each hypothesis. Additionally, we provide reasoning for each set of hypotheses and predictions.

Pg 5-6, Lines 105-130: In wild barn swallows (Hirundo rustica erythrogaster), we asked three questions and explored related hypotheses about the effects of near nest temperature on nestling growth. First, we asked: Does the effect of temperature on nestling mass differ when the exposure is assessed during early versus late development? Given that temperature exposures during different stages of development differentially effect nestling metabolic rate and thermoregulatory strategies (e.g., [5,6,15]), we hypothesized that due to differences in metabolic costs, exposure to extreme and variable temperatures early in development would have a stronger effect on nestling mass than exposure later in development. We predicted that hot, cold, and variable temperatures from hatch to day five (early in development), before expected development of thermoregulatory abilities, would have stronger negative effects on nestling mass before fledging than temperatures from day six to twelve (late in development) (Fig 1).

Our second and third questions asked whether social experiences modify the effect temperature on nestling growth. For question two, we asked: Does the effect of temperature on nestling mass depend on whether a nestling is the smallest in the brood? For question three, we asked: Does the effect of temperature on nestling mass depend on the amount of feeding provided by parents? Differences in nestling size are associated with heterogeneity in access to resources [21–23] and can impact thermoregulatory ability [6,24]. In addition, previous studies report that parental care can buffer nestlings from effects of adverse conditions [19,31]. Therefore, we hypothesized that disadvantageous social environments (e.g., being the smallest in the brood, lower levels of parental feeding) would exacerbate the negative impacts of extreme and variable temperatures on nestling mass due to the nestlings’ reduced access to resources. We predicted that hot, cold, and variable temperatures would have stronger effects on nestling mass for the smallest nestling in the brood relative to other brood mates and that temperature exposure would have stronger effects on nestling mass at low than high levels of parental feeding (Fig 1). We set out to test these hypotheses in an exploratory nature, as a step toward understanding how the developmental and social conditions under study may shape temperature effects on nestlings.

Comment. L103. I would suggest just one figure that has the multiple panels be in main body of ms. No need for this sentence explaining that information.

Response. Done. Because these conceptual figures include the variables under study and are relevant to the specific linear mixed models we ran, we reference this new figure (Fig 3) in the “Statistical analysis” subsection of the methods.

Comment. L107. Cut this sentence from the figure legend.

Response. Done.

Comment. L114. One limitation of this work in relation to climate change is that you are studying only a single year. I would suggest that the key part of your work is really on intra-annual variability as that was all you are really going to be able to do.

---

## [Editor Report · Decision Letter 2]

16 Mar 2026

The effects of temperature on nestling growth in a songbird depend on developmental constraints

PONE-D-25-53652R2

Dear Dr. Madden,

We’re pleased to inform you that your manuscript has been judged scientifically suitable for publication and will be formally accepted for publication once it meets all outstanding technical requirements.

Kind regards,

Christopher A. Lepczyk

Academic Editor

PLOS One

Additional Editor Comments (optional):

I appreciate the clear revisions to the manuscript. The work is very interesting and adds to our knowledge of how development relates to a changing climate.
---

## [Editor Report · Acceptance letter]

PONE-D-25-53652R2

PLOS One

Dear Dr. Madden,

I'm pleased to inform you that your manuscript has been deemed suitable for publication in PLOS One. Congratulations! Your manuscript is now being handed over to our production team.

Kind regards,

on behalf of

Dr. Christopher A. Lepczyk

Academic Editor

PLOS One